# An Evaluation of the Efficacy of Very High Resolution Air-Quality Modelling over the Athabasca Oil Sands Region, Alberta, Canada

**Matthew Russell[1], Amir Hakami[1], Paul A. Makar[2], Ayodeji Akingunola[2], Junhua Zhang[2], Michael D. Moran[2], and Qiong Zheng[2]**

[1]Department of Civil and Environmental Engineering, Carleton University, Ottawa, Canada

[2]Air Quality Research Division, Environment and Climate Change Canada, Toronto, Canada

## Abstract

We examine the potential benefits of very high resolution for air-quality forecast simulations using a nested system of the Global Environmental Multiscale – Modelling Air-quality and Chemistry chemical transport model. We focus on simulations at 1km and 2.5km grid-cell spacing for the same time period and domain (the industrial emissions region of the Athabasca Oil Sands). Standard grid cell to observation station pair analyses show no benefit to the higher resolution simulation (and a degradation of performance for most metrics using this standard form of evaluation). However, when the evaluation methodology is modified, to include a search over equivalent representative regions surrounding the observation locations for the closest fit to the observations, the model simulation with the smaller grid cell size had the better performance. While other sources of model error thus dominate net performance at these two resolutions, obscuring the potential benefits of higher resolution modelling for forecasting purposes, the higher resolution simulation shows promise in terms of better aiding localized chemical analysis of pollutant plumes, through better representation of plume maxima.

## 1   Introduction

Numerical modeling of the atmosphere in an Eulerian framework relies on discretization of the computational domain into a numerical grid. The horizontal grid cell size of atmospheric simulations can range from hundreds of kilometers, to the metre-scale of Large Eddy Simulation models. Air-quality model grid-cell size typically follows the grid-cell sizes used in weather forecasting models, which in turn have followed a gradual progression towards finer discretization where more explicit representation of cloud formation and local radiative transfer effects may be represented. The most recent weather forecasting applications (e.g. Leroyer *et al.*, 2014) have reached grid-cell sizes as small as 250m over limited domains such as individual cities, and have shown promising results in terms of being able to resolve some aspects of local circulation. In addition, as grid resolution reaches the 3 to 4 km scale, explicit cloud microphysics packages may be used, allowing potentially better performance, particularly with regards to feedbacks between meteorology and chemistry (Yu *et al.*, 2014; Gong *et al.*, 2015). However, while these models promise better physical representation of local chemistry, their performance may be limited by the quantity and availability of initialization and boundary condition meteorological data; these data may be used in a data assimilation context to improve their initial state. The accuracy of broader-scale meteorological

predictions may thus influence local model accuracy, despite the ongoing decrease in meteorological model (and
consequently air-quality model) grid cell size.  Some recent air-quality model simulation studies with grid cell sizes
on the order of one to four km include Thompson and Selin (2012), Li *et al.* (2014), Joe *et al.* (2014), Kheirbek *et al.*
(2014), Kheirbek *et al.* (2016), and Pan *et al.*, (2017).
For the purposes of this study, Very High Resolution (VHR) modelling refers to the current higher resolution limits
of chemical transport models (CTMs), employing a horizontal grid cell spacing of 1km or less.  It is in this regime
that the photochemical processes may be forecasted with resolved microphysics (e.g. Milbrandt and Yau,
2005(a,b)), and detailed particle and gas-phase chemistry, using currently available computer technology. VHR
modelling is very computationally expensive, and also introduces its own set of challenges, such as the availability
of surface boundary condition fields as the model grid cell size decreases. Moreover, it is not currently clear
whether decreases in model grid cell size leads to more accurate results when compared to observations. The
motivation behind VHR modelling in CTMs is to reduce the impact of diluting chemical concentrations - especially
from averaging emission plumes into large grid cells – in order to better capture inhomogeneities in emission
profiles, to better simulate local transport processes associated with terrain that would otherwise be smoothed by
the use of a coarse grid, and to reduce truncation errors and hence achieve better numerical accuracy (Jacobson,

49    1999).

We note here that while the terms "grid cell size" and "resolution" tend to be used interchangeably in the
literature, this is not true in a precise mathematical sense; more formally, the ability to resolve features of size
$2\Delta x$ requires a grid cell spacing of size $\Delta x$, and the highest spatial frequency which can be reconstructed from a
discrete sampling of the latter grid cell spacing will be $\frac{1}{2\Delta x}$, the Nyquist wavenumber of the grid cell size
discretization.  Furthermore, atmospheric models may make use of energy dissipation techniques that broaden
the size of resolvable wavelengths to $3\Delta x$ to $4\Delta x$ (Grasso, 2000; Pielke, 2001).  Model resolution is thus a function
of, but not equivalent to, grid cell size. Here, we define "resolution" as the ability of a model to clearly distinguish
components of a predicted atmospheric variable, as a *function* of grid cell size.
The issue of a model to distinguish these features is also compounded by uncertainties in model inputs. For
example, in a large rural setting, a large model grid cell will represent an area containing many roads, whose
emissions will be averaged into one value per species per time. As the grid cell size decreases however, this
averaging effect will be reduced, giving each road's emissions more impact on the resulting concentrations in the
grid cell containing it. However, the smaller grid cell size will also result in steeper concentration gradients in the
model between adjacent grid cells, which can in turn result in numerical instabilities that contaminate predictions
(Salvador et al., 1999).  At the same time, a reduction in grid-cell size can be shown formally to reduce
inaccuracies in the discretization of the governing equations for atmospheric motion (Coiffier, 2011).  Previous
efforts to address these issues through variable grid size or structure in air quality modeling have not received
sustained attention, and therefore most current air quality models use a uniform (albeit nested) grid cell size in
applications (Garcia-Menendez *et al.*, 2010; Kumar *et al.*, 1997).
As resolution increases further, the presence of local topographical features (*e.g.* buildings and street canyons)
become more important. Both the increased topographic complexity, and potential numerical instabilities can
lead to differences in meteorological forcing as resolution increases (Wolke, *et al.*, 2012; Gego, *et al.*, 2005)). The
contribution of meteorological uncertainties due to resolution become more significant, especially for secondary
pollutants such as ozone (Valari and Menut, 2008) or secondary Particulate Matter (PM). For example, Markakis *et*
*al.* (2015) in their analysis of 4 km CHIMERE simulations for the relatively flat terrain of Paris, France, suggested
that model meteorological grid cell size does not significantly impact forecast accuracy. That may not have been
the case, had their terrain been more complex. In contrast, Queen and Zhang (2008) observed considerable
meteorological sensitivity to the more complex terrain in their 4 km resolution Community Multiscale Air Quality
(CMAQ, EPA 1999) model simulations over the Appalachian Mountains in the eastern United States, as did
Salvador *et al.* (1999) for meteorological model simulations.
A number of studies have tried to evaluate the benefits of higher resolution simulations and to quantify the
impact of sub-grid variability by using different model grid-cell sizes  (Vardoulakis *et al.*, 2003; Ching *et al.*, 2006;
Pepe *et al.*, 2016).  These studies have often demonstrated that failure to account for higher resolution features
may result in mischaracterization of concentrations or health impacts (Isakov *et al.*, 2007), although the capability
of current models to provide this information with sufficient accuracy is unclear.  One study found that increasing
resolution did not change predicted health outcomes, and concluded that "resolution requirements should be
assessed on a case-by-case basis" (Thompson and Selin, 2012), while others (e.g. Kheirbek *et al.* (2014), Kheirbek
*et al.* (2016)) have employed 1km resolution without discussing the impacts of resolution on predicted health
outcomes.   Population exposure studies using air pollution models may be affected by resolution in a more
complex fashion, given that both the predicted field (a pollutant with a known health impact) and the data to
which the predicted field is to be linked (the human population) both have resolution dependencies.  The health
studies carried out to date highlight the need for better understanding the underlying controlling factors for
model accuracy with decreasing grid cell size.
Terrain and meteorology are not the only factors that contribute to greater uncertainties as horizontal grid cell
size is reduced – for example, the ability of the model to locally resolve emission fluxes may also become a factor.
This may result in improved or deteriorated model performance as the size of the grid cells decrease. Gridded
model emissions may have an intrinsic resolution dependence in the underlying spatial disaggregation fields, and
this can contribute to uncertainties and errors in emissions as grid cell size is decreased. For instance, Valari and
Menut (2008) found that the discrepancy between their modelled and observed concentrations grew, rather than
shrank, in response to decreases in grid cell size from 48km to 6 km, and they associated these results with

changes in the resulting local emission fluxes. They showed that in their model setup, with regard to ozone, a grid cell size was reached ($12 \times 12$ km$^2$) where errors in inputs (errors in the emission inventory, wind direction, *etc.*) outweighed the importance of other sources of model error such as grid cell size. The authors however noted that Paris' ozone photochemistry very often resides on the transition between a $NO_x^-$ sensitive and a VOC–sensitive regime (Sillman *et al.*, 2003). These are chemical conditions which can alternatively produce or titrate ozone, and hence have a degree of sensitivity to precursor emissions, and therefore, also, to any errors in those emissions. Conversely, in a 3-level nested 9- to 3- to 1- km MM5–CMAQ simulation over Osaka, Japan, Shrestha *et al.*, (2009) found that ozone comparisons to observations improved as the grid resolution increased. This was also the case for a 36- to 12- to 4-km nested MM5–CMAQ simulation over Houston, USA (Ching *et al*., 2006), where the ozone forecast improvement associated with higher resolution was attributed to the ability of the finer grid cell size model nests to adequately resolve high concentrations of freshly emitted NOx and hence allow for more local ozone titration. The latter process might not take effect until the grid cell size is sufficiently fine to resolve the $NO_x$ source patterns (*i.e.,* a level where traffic and industrial sources can be identified.) This titration was not seen until they decreased their grid cell sizes to 2 km and smaller. Stroud *et al.* (2011) noted a similar grid cell size dependent chemical impact on model performance, where secondary organic aerosol formation maxima were better simulated with a 2.5km grid cell size model than a 10km grid cell size model. In general, the impact of resolution on model performance appears to depend on a number of factors, such as the terrain, spatial distribution of sources, pollutant of concern, season, *etc*. (Arunachalam *et al.,* 2006; Queen and Zhang, 2008; Dore *et al.*, 2012).

Salvador *et al*. (1999) studied the prediction accuracy impacts of meteorological model grid cell size in a region with complex domain, and found that 2km or smaller grid cell sizes were required to resolve local scale complex terrain flow features, and that daytime vertical advection and predictions of turbulent kinetic energy and potential temperature were influenced by grid cell size. Dore *et al*. (2012) evaluated air quality model $NO_2$ simulations employing 1, 5 and 50km grid cell sizes against observations, and found the best performance for the 1km simulation, with more physically realistic distributions of reactive nitrogen, attributing this performance gain to more realistically precipitation simulations and emissions inputs for the smallest grid cell size. The availability of high-resolution emissions information may be a limiting factor in improved simulations as grid cell size decreases. Valari and Menut (2008) noted that emissions inaccuracy was the principal cause of noise in small grid cell size simulations conducted for the Paris area, and proposed the use of statistical downscaling in favour of predictive modelling at scales at or below 1km grid cell size. The current state of model science is typically evaluated through multi-model intercomparisons (e.g. Im *et al*., 2015), and the meta-analysis of these studies can be used to provide useful benchmarks to assess current model performance for specific model species and observations (Emery *et al*., 2017). However, such studies do not identify the causes for good or poor performance relative to the benchmarks – diagnostic studies, "in which chemical and physical processes within the model are analyzed

individually and collectively" (Emery *et al.*, 2017) are required for this purpose.    Examinations of the impact of
model grid cell size on performance are an example of such a diagnostic evaluation.
The benefits for model performance with increased spatial resolution are unclear, based on the above literature.
However, most papers converge towards the following qualitative conclusions:

1. The impact of terrain topology on meteorological forcing as grid cell size decreases can dwarf the impact of
a more accurate spatial apportionment of the corresponding emissions.

2. Decreases in grid cell size result in a more realistic spatial distribution of chemical species, whether or not
model performance is improved.

3. Uncertainties of spatial and temporal emissions allocation have an increasing influence on overall model
uncertainty as model grid cell size decreases.

The 1980's saw several studies in which the potential impacts of wind direction errors on dispersion model
performance were examined. Fox (1981) noted that pairing of model output at observation station locations could
be done as a function of both time and space: as a function of time (by combining the data across all stations), as a
function of space (by combining all times, at each station location), or without any pairing (observations and data
were compared as cumulative frequency distributions). The accuracy of regulatory dispersion models in the early
1980's was such that Fox (1984) concluded that model and observation values paired in time and space exhibited
"little to no correlation" and discussed potential errors associated with transport. Poor correlations were also
noted by Hanha (1988), reporting on the first generation of reactive-transport models, stated "wind direction errors
are the major cause of the poor agreement in hourly predictions of concentrations at short distances downwind of
point sources," as well as describing metrics for air-quality model evaluation. Hanha (1988) also noted that model
predictions could be offset in space and time relative to observations, leading to poor performance statistics,
despite a greater degree of similarity of behavior if the offsets are taken into account. Errors in wind-field
modelling were described as the main source of error in simulations of plumes by Carhart *et al* (1989), again
showing how better agreement resulted when model and observations were unpaired in time and/or space, and
noted that other metrics such as maximum plume width might better represent model performance. Lee (1987)
found that small perturbations in space and time could result in poor correlations, despite similar histogram
distributions of both model and observations.
More recently, Kang *et al.,* (2007) examined the concept of using the area of the limiting resolution of the model (2
to 3$\Delta$x, where $\Delta$x is the horizontal grid cell size) to weight or spatially average model evaluation metrics for a single
grid-cell size, noting how the model's rated ability to capture high concentration events ("hits") was increased when
the limiting resolution of the model was incorporated into the performance metrics. However, the use of averaging
may mask the potential for a model with a small grid cell size to contain both the desired plume magnitude, as well
as much lower concentrations, within the same larger representative area, in turn masking the potential impact of
the reduction in grid cell size.
We expand on this concept to evaluate the impact of model grid cell size in the context of an equivalent area about
a given observation location.  We examine area-weighted metrics in the form of averages over roughly equivalent
areas for different model grid cell sizes, and also use the *a priori* knowledge of the observations to determine
whether the closest match to observations may be found within an equivalent area. We show that the latter metric
demonstrates a positive impact of model grid cell size on simulation results, while more simple paired comparisons,
and averages over similar areas, mask these benefits.
We examine the impact of grid cell size on model performance in a region of intense petrochemical extraction and
upgrading, the Athabasca Oil Sands Region (AOSR). The AOSR refers to the northernmost of three large bitumen
deposits located the northern part of the province of Alberta in Canada; the Athabasca, Peace River, and Cold Lake
areas. Together these areas cover 142,200 km$^2$ in total, and constitute the third largest oil reserves in the world
(Government of Alberta, 2016), as shown in Figure 1.  The oil sands sector is the second largest source of $SO_2$ and
the third largest source of industrial $NO_x$ in the province of Alberta. This sector is also a significant source of
industrial PM, CO, and Volatile Organic Compound (VOC) emissions (Zhang *et al*., 2018), from a variety of source
types and industrial processes (*e.g.* open pit mine tailings ponds, large diesel fleets, bitumen upgrading facilities).
As is described below, very high resolution emissions data are available for these sources, and emissions take place
in a region with significant topography, hence the region provides a good test case for the relative impact of grid
cell size on air-quality model prediction results.
We describe next our model, the simulation domains and forecasting setup, the emissions data, our evaluation
methodology, and the results of our analysis.

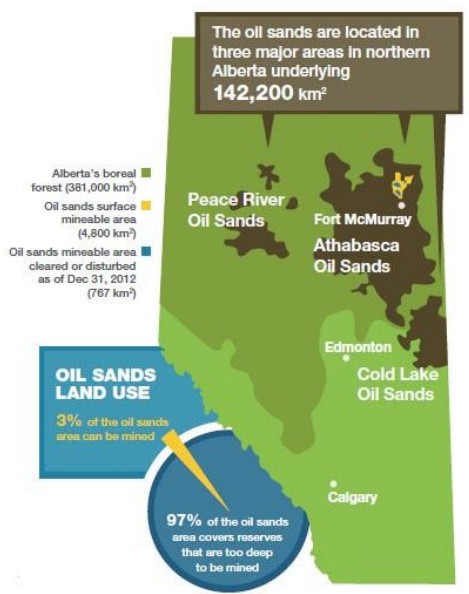


Figure 1.  Map showing the Oil Sands regions (Government of Alberta, 2016).

## 2. Methodology

### 1.1   GEM-MACH

The air-quality model used in this work is Environment and Climate Change Canada's (ECCC) Global Environmental Multiscale – Modelling Air-quality and Chemistry (GEM-MACH) model, which has been in use as Canada's operational air-quality forecast model since 2009 (Moran *et al.*, 2010).  GEM-MACH is an on-line model, that is, both meteorological and chemistry processes are handled within a single model.  The chemical processes reside within the physics module of the Global Environmental Multiscale meteorological forecast model (Côté, *et al*., 1998(a,b)), originate with Environment Canada's earlier off-line model (A Unified Regional Air-quality Modelling System; AURAMS, Gong *et al.*, 2006), and include process representation for particle microphysics (Gong *et al*., 2003(a,b)), inorganic heterogeneous chemistry (Makar *et al.*, 2003), aqueous phase chemistry, in-cloud and below-cloud scavenging (Gong *et al.*, 2006), and secondary organic aerosol formation (Stroud *et al*, 2011).  GEM-MACH employs a sectional approach to represent the size distribution of atmospheric particles, with 12-bin (Makar *et al.,* 2015(a,b); Gong *et al.,* 2015) or 2-bin configurations (Moran *et al.*, 2010).   The latter configuration is designed for maximum computational efficiency, with re-binning to the 12-bin distribution for key particle microphysics processes, in order to improve accuracy.  Here, the 2-bin version of the model has been used, the main focus of the work being the impact of horizontal grid cell size on model results.  Eight aerosol chemical components are resolved in GEM-MACH (sulphate, nitrate, ammonium, elemental carbon, primary organic aerosol, secondary organic aerosol, sea-salt and crustal material).  In the present study, we make use of GEM-MACH v.1.5.1, described in more detail in Makar *et al*., 2015(a,b), employing 80 levels in a hybrid vertical coordinate system extending up to 0.1hPa (~30km).   Both model grid cell size simulations compared here (2.5km and 1km grid cell sizes, see below) make use of the Milbrandt-Yau double moment explicit microphysics scheme, that is, cloud processes are resolved explicitly at these scales (Milbrandt and Yau, 2005(a,b)).

### 1.2    Model Setup

#### 1.2.1    Grid Nesting

Four levels of nesting have been employed in our simulations, shown in Figure 2(a).  This version of GEM-MACH operates on a rotated latitude-longitude coordinate system wherein the position of the coordinate system poles is set by the user, allowing rotations of the grid with decreasing grid cell size during nesting.  The outermost nested grid corresponds to the westernmost two-thirds of the operational GEM-MACH forecasting domain, with a 10km grid cell size, and employ a combination of the Kain-Fritsch sub-gridscale convective cloud scheme (Kain and Fritsch, 1990; Kain, 2004) and a Sunqvist (1988) for cloud parameterizations.  Within that outer grid is nested a 10 km grid cell size western Canada domain (yellow region, Figure 2(a)) which has been rotated to match the horizontal orientation of the Rocky Mountains, and which makes use of a  double-moment microphysics scheme (Milbrandt  and  Yau,  2005a,b)  in  place  of  the  Sundqvist  (1988)  parameterization.   The  intention  of  this

intermediate local 10km simulation domain was to provide initial hydrometeors for the two innermost domains,
to reduce the "spin-up" time required for the inner domains' meteorology to reach an equilibrium with respect to
cloud formation.  The latter two domains (2.5km and 1km grid cell sizes) resolve the cloud microphysics explicitly
using the double moment scheme alone and no convective parameterization (Milbrandt and Yau, 2005a,b).   The
third nested grid inwards (green region, Figure 2(a)) is the 2.5km grid cell size domain, which covers most of the
Canadian provinces of Alberta and Saskatchewan.  This grid will hereafter be referred to as the OS2.5km domain.
The fourth and final nested grid (blue square, Figure 2(a)) is a 1km grid cell size domain, roughly centered over and
covering the immediate environs of the Athabasca Oil Sands, and is referred to hereafter as the OS1km model.
This last nest also shows the region within which 22 instrumented aircraft flights were conducted during August
and September of 2013, providing a unique measurement dataset for our evaluation of the OS2.5km and OS1km
model output for the same time period. Table 1 provides details on the horizontal dimensions of each of these
nested domains, and the duration of the simulations on each grid.  All four model nests make use of the same
vertical coordinate and levels.  Figure 2(b) shows the topography of the 1km domain in detail; the region to be
modelled is situated in a broad river valley, with a local vertical relief of 750 m.  Significant wind shears and
frequent inversions are observed in the region, and part of our interest in 1km grid cell size simulations is to
determine the extent to which these local features may influence model prediction accuracy.
2.2.2 Simulation Cycling Strategy
Model simulations mimic an operational forecasting system, starting from the use of archived, data-assimilated
meteorological analyses as meteorological input and boundary conditions every 36 hours.   The use of analysis
fields is a standard meteorological forecasting practice to prevent the chaotic drift of the model results from
observed meteorology over time.  The outermost 10km domain uses initial and boundary conditions from the
output of a meteorological simulation, that is itself driven by an analysis field.  The outermost domain model then
carries out a 36-hour forecast, of which the first 6 hours are discarded as spin-up; the final 30 hours are used as
initial and boundary conditions for the rotated 10 km grid cell size domain (the OS10km domain).   An OS10km
simulation of 30 hours is then carried out, with the first 6 hours being discarded as spin-up, and the latter 24 hours
forming the initial and boundary conditions for the 2.5 km grid cell size OS2.5km simulation. The OS2.5km
simulation is of 24 hours duration.  The OS1km simulation covers the same 24 hours (and hence both 2.5km and
1km simulations start from the same OS10km initial conditions at for every 24 hour forecast), with the 2.5km
simulation providing boundary conditions thereafter to the OS1km model. Continuity between 24 hour forecasts
is thus maintained at the level of the outermost nest.  The outermost domain is cycled every 12 hours starting at
0UT and 12UT; however, we have selected the set of contiguous OS2.5km and OS1km 24 hour simulations starting
from the 12UT continental domain for our comparison.
Meteorological boundary conditions for lowest resolution GEM-MACH simulations are taken from operational
GEM forecasts, in turn driven by data assimilation analyses performed at the Canadian Meteorological Centre.

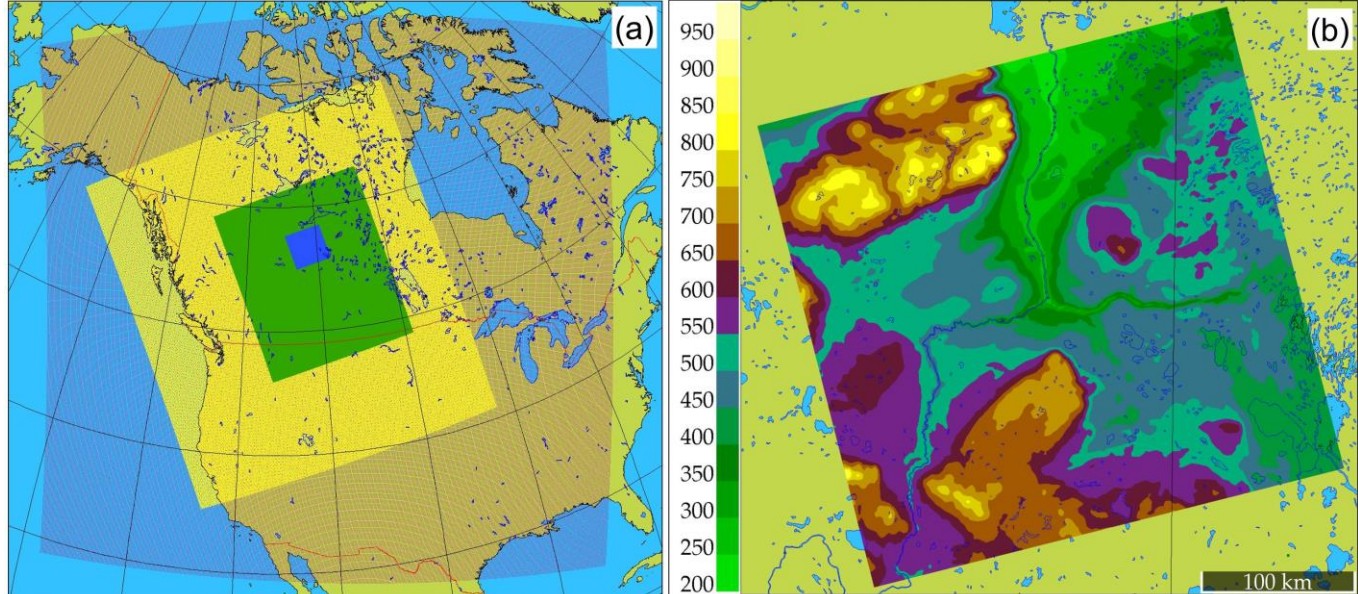


Figure 2. (a) The four nested domains of the GEM-MACH simulations.  From outermost to innermost domains,
these are CONT10km (outermost, red dots), OS10km (yellow), OS2.5km (green), and OS1km (blue).  The model
simulations from the two innermost domains are the focus of the present study. (b) Topography in the OS1km
domain centred on Fort McMurray, Alberta (m agl).  The coloured area corresponds to the central blue domain in
(a).
Table 1.  Nested Domain Specifications

| Parameter | CONT10km | OS10km | OS2.5km | OS1km |
|---|---|---|---|---|
| Grid Size | 520x520 | 318x280 | 643x544 | 318x324 |
| Time step size (s) | 300 | 300 | 60 | 20 |
| Hours simulated | 36 | 30 | 24* | 24* |

*Note that both OS2.5km and OS1km output frequency was hourly.
2.3  Model Emissions
All emissions data used in this work are described in Zhang *et al*. (2018).  These emissions data include (a) direct
observations of stack-specific hourly emissions measured by Continuous Emission Monitoring Systems (CEMS), (b)
regional emissions inventory data from the Cumulative Environmental Management Association (CEMA) - which
had the most detailed  stack and process level emission data for the AOSR facilities, including emissions from mine
faces, tailings ponds, and the off-road mining fleet), (c) the 2010 Canadian Air Pollutant Emissions Inventory (APEI)
-  which is the most comprehensive national emissions inventory, and which has the largest spatial coverage for
area sources outside the AOSR, and (d) the 2013 National Pollutant Release Inventory (NPRI) (a subset of the APEI)
that is based on emissions reports from large industrial facilities.
These emissions data sets primarily describe emissions of pollutants known as criteria-air-contaminants ($NO_x$,
VOCs, $SO_2$, $NH_3$, CO, $PM_{2.5}$, and $PM_{10}$) for *major-point sources* (*i.e.*, large emission stacks) and *area sources*.  Area
emissions sources typically consist of multiple small mobile sources spread over a large area (*e.g.,* off-road
vehicles), large flux sources such as mine tailings settling ponds or mine faces, and/or large numbers of small
stacks for which no stack characteristic data (volume flow rates, temperatures of emissions, stack diameters),
needed to estimate plume-rise heights, are available.
Major-point sources are represented by a single geographical (latitude, longitude) pair of coordinates, and are
assigned to the grid cell in which the point is located.  These sources are likely to be the most impacted by model
horizontal grid cell size, as even a large major-point source plume, which in reality may only occupy an emissions
horizontal area on the order of 100 $m^2$, is represented by a flux spread over an entire grid cell.  A plume from a
major point source within a 2.5km grid cell will thus be immediately diluted to a size of 6.25$km^2$ upon emission,
whereas the same source with a 1km grid cell will have a cross-sectional horizontal extent of 1$km^2$.  At the same
time, higher resolution may require a much more accurate representation of model winds close to the sources to
maintain accuracy in evaluation metrics dependant on plume position such as correlation – a wider plume being
more likely to at least partially intersect a monitoring station location than a narrower plume.
Area sources that are large compared to both model grid cell sizes (2.5km and 1km) can be expected to be
approximated by model grid cells of both resolutions, and are thus expected to be less impacted by model
resolution than emissions from point sources.  However, smaller area sources (*i.e.* areas intermediate between
2.5km and 1km to the side) may be better resolved, and hence have less dilution and higher downwind
concentrations, when higher spatial resolution is employed.
In the AOSR, approximately 95% of the $SO_2$ emissions originate in major-point sources, while $NO_2$ is
approportioned ~40% to major-point sources and ~60% to area sources (Zhang *et al.*, 2018).  Consequently our *a
priori* expectation is that the impact of the resolution change will be strongest for species like $SO_2$, and less strong
for species like $NO_2$ that are emitted in part by point sources, but may also be apparent for other species and
secondary products, such as $O_3$.
1.4    Model Evaluation Methodology and Metrics
Comparisons between air-quality models and observations usually take the approach of comparing observation
and model-generated values paired in time and space, from the observation location and corresponding model
grid-cell respectively. We refer to this approach hereafter as our "standard" evaluation, for both 2.5km and 1km
simulations. However, we note additional factors aside from grid-cell size may influence the outcome of air-
quality model evaluations. For example, the relative skill of the meteorological component of the air-quality
model will depend in part on the density of meteorological observation data, incorporated into the model via data
assimilation, for the construction of the model's initial meteorological state. This in turn will influence the local
skill of the model's predicted wind directions and hence the skill of its plume transport. The simulations carried
out here focus on the Fort McMurray area, where the nearest available upper air meteorological sounding site is
located at the ECCC Stony Plain station, located approximately 500km south-west of the study area. The
advantage of higher resolution simulations (*e.g.*, reduced numerical error associated with the discretization of
transport operators, and better treatment of local topographic influences) may thus be offset by errors in the
predicted *large scale* flow.
While meteorological model synoptic-scale forecast errors may manifest themselves locally as errors in the
direction of winds driving local plume transport, other advantages may result from the use of higher resolution
air-quality models. Since lower resolution models *de facto* instantaneously redistribute plumes emitted from
large stack sources over a larger area, such artificial diffusion will reduce the model's ability to accurately simulate
concentration maxima, and the resulting chemistry, within simulated model plumes. However, the spatial extent
of a plume in a model employing a large horizontal grid cell size may be such that its existence may be captured at
discrete observing sites. In contrast, forecast plumes in models with smaller horizontal grid cell sizes may
correctly capture plume magnitude and chemical behaviour, but may be more subject to errors in the larger scale
wind direction. To illustrate this point, Figure 3 shows a conceptual diagram of an actual plume, a large grid cell
size model plume, and a small grid cell size model plume, where the latter two simulated plumes are both subject
to the same synoptic-scale error in wind forecast direction (indicated by large red arrows; the smaller red arrow in
Figure 3(c) indicates the impact of local forcing predicted for the second model). Observation station "+A" is
located downwind, and records the presence of the actual plume (Figure 3(a)). The coarse grid cell size simulated
plume (Figure 3(b)), despite the error in the forecast wind direction, captures part of the observed plume in the
resulting time series at the observation station location. In contrast, the small grid cell size plume (Figure 3(c)),
despite resolving the plume shape (and plume-internal chemistry) to a greater degree than the coarse grid cell size
simulated plume, fails to record the presence of the plume at the observation location. A simple paired
observation-model time series evaluation would thus suggest that the former model has superior performance to
the latter model in this example, despite the latter model having created a more "realistic" plume in terms of the
maximum concentration reached, albeit in the wrong location, due to synoptic-scale forecast wind direction error.
In this particular instance, the magnitude of the smaller grid cell size simulated plume is more realistic than that of
the coarse grid cell size plume, but this improvement will not be captured in a standard evaluation analysis. Shifts
in plume location across individual grid cells away from the location of an *in-situ* observation are more likely grid
cell size decreases. In this example, a standard analysis would impose a more stringent expectation on the smaller
grid cell size simulation to correctly identify plume locations.

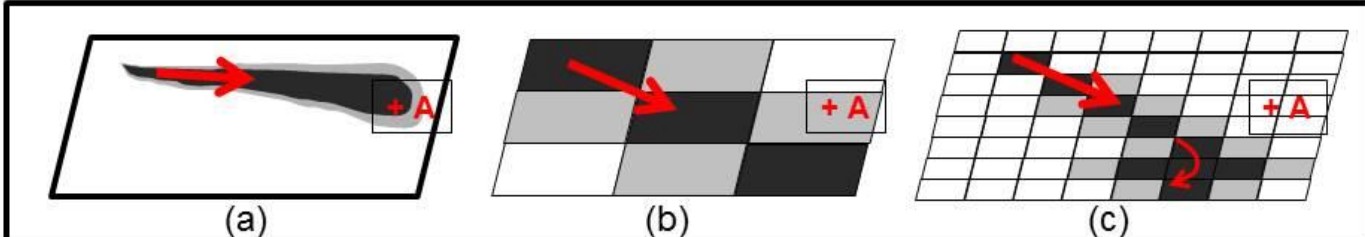

Figure 3. Schematic comparison of surface concentration contours and model grid cell values of a transported pollutant
plume from a large stack (termed a "point" source). Wind direction shown by red arrows. Monitoring station location
marked by "+A". (a) Actual plume. (b) Coarse grid cell size air-quality model prediction. (c) Fine grid cell size air-quality model
prediction. Note the change in wind direction between observations (a) and simulations (b,c) associated with errors in the
forecast of the synoptic wind.
In addition to the standard analysis, we perform additional analyses that examine the model's ability to resolve
plumes in the *vicinity* of the observation station, in order to attempt to evaluate the potential for higher
resolution simulations to provide benefits which may be masked by synoptic scale forcing errors. This strategy is
illustrated in Figure 4.

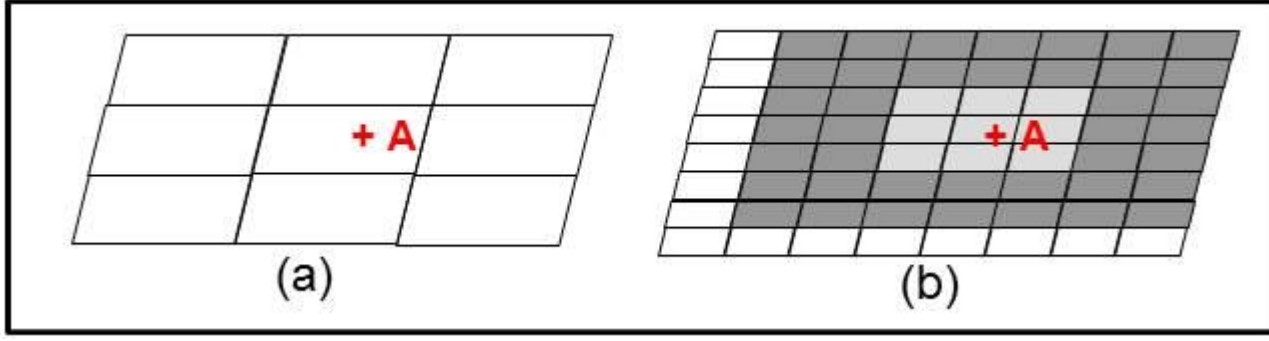


Figure 4. Scale diagram of the same region in (a) 2.5km grid cell size simulation and (b) a 1km grid cell size simulation.
Region enclosed by light grey / dark grey shading in (b) represents the nearest nine / forty-nine 1km gridpoints surrounding
the observation location "A".
Figure 4(a) shows an observation station enclosing the nine nearest-neighbour model grid-cells for a 2.5km grid
cell size, while Figure 4(b) shows the corresponding 1 km grid cell size map, with the nine nearest-neighbour
model grid-cells shown in light grey, the forty-nine nearest grid cells shown in the region enclosed in dark grey.
Figure 4(a) encloses a region of 56.25 km$^2$ (7.5x7.5 km), while the light grey region in Figure 4(b) encloses 9km$^2$,
and the darker grey region encloses 49 km$^2$.
As noted above, in a formal mathematical sense, the smallest region resolvable by an Eulerian grid model is twice
the size of the model grid cell size (relating to the Nyquist frequency of the model); hence the smallest resolvable
feature spans two model grid cells in each direction. However, in a practical sense, a total of nine grid cells

centred on the observation station must be used to allow a boundary of two grid cells in any direction. Sampling any or all of the 9 grid cells in Figure 4(a) may thus be said to be representative of the model's ability to simulate events occurring at discrete location "+A". The closest corresponding sampling region available to the 1 km model (Figure 4(b)) is shown in dark grey. The light grey region of Figure 4(b) represents the closest 1 km grid cell size region that corresponds to the single 2.5 km grid cell in which the observation station is located in Figure 4(a). We attempt to ascertain model performance in these approximately equivalent regions around each observation station, in the analysis that follows.

Our approach follows two steps:

(1) From the 2.5km simulation, in addition to the predicted model value at the grid-cell containing the observation location, we determine the model grid-cell value in the nine grid-cells surrounding the observation station location which has the closest value to that observed at the station. This represents the model's "best estimate" of the value at the observation station location itself, to the model's ability to resolve features at 2.5km grid cell size.

(2) From the 1km simulation, in addition to the model value at the grid-cell location, we select the closest value to the observation value from: (a) the nearest nine grid-cells to the observation station location, and (b) the nearest 49 grid-cells to the observation station location. The former represents the model's "best estimate" of the value at the observation station location itself, while the latter represents the 1km model's best estimate in the closest equivalent region to the limiting resolution of the 2.5km model.

Comparing the resulting statistical measures of each of these selected values with observations, in addition to the standard analysis, thus evaluates the model's best attempt to resolve features for the specified grid cell size, and allows cross-comparison of model performance within nearly equivalent areas. Cross-comparing the statistical values for the different regions described above shows the model's ability to resolve features such as plumes from the standpoint of the region represented at the different grid cell sizes. If synoptic-scale transport direction errors creates situations similar to that depicted in Figure 3(a), a standard comparison of error would be expected to show little benefit to higher resolution. However, the "best model estimate" comparisons would capture the ability of the higher resolution model to more accurately simulate the magnitude of the plume, if not its spatial location. Each of these selection procedures will be employed in the surface concentration comparisons which follow.

We evaluate our model simulations against observations made at surface monitoring networks in the vicinity of the Athabasca oil sands, and aboard an instrumented aircraft, the National Research Council of Canada Convair. For the surface monitoring data, hourly time series of model output were matched to station time series using the different strategies described above. For the aircraft observations, we extract model values through temporal and

spatial interpolation to the aircraft's position during the flights and only perform the standard analysis, as well as
examining the behaviour of the two simulations along cross-sections corresponding to the flight paths.
Our statistical metrics for evaluation are common to many other air-quality applications, and were computed
using the 'modstat' function from the OpenAir R package (Carslaw and Ropkins, 2012). Further discussion of
different metrics for model evaluation may also be found in Yu *et al.,* (2006). The statistics calculated here
include: mean bias (MB; perfect score: zero), mean absolute gross error (MGE; perfect score: zero), normalised
mean bias (NMB; perfect score: zero), normalised mean gross error (NMGE: perfect score: zero), root mean
squared error (RMSE; perfect score: zero), correlation coefficient (r, perfect score: unity), coefficient of
efficiency (COE: a perfect score is unity, a zero/negative score means the model is equivalent/less predictive
than the mean of the observations), and the index of agreement (IoA; perfect agreement is unity, and -1
indicates no agreement or little variability).

## 2 Simulation Comparisons and Evaluation

### 3.1 Model-to-model comparisons and averages

We begin a comparison of 2.5km and 1km grid cell size for specific events, and for averages across the 1km
domain, in order to provide a qualitative comparison of the differences in simulations for the two simulations, and
then continue with the quantitative comparison. Figure 5 compares OS2.5km (left column) and OS1km (right
column) simulation results for a cross-section located 0.2km from a major $SO_2$ emissions source at 0, 12 and 24
hours into a given simulation day.
The model results are identical at hour 0 due to both the OS2.5km and OS1km models being initialized from the
OS10km data at this time (small differences in Figure5(a,b) are due to slight mis-matches in the cross-section
locations). Subsequent cross-sections show the OS1km model is capable of resolving both higher absolute mixing
ratio values, and sharper gradients, within 12 hours of simulation time (Figure 5 (c,d)). Multiple plumes are
resolved by 12 hours of simulation time in the 1km grid cell size simulation, along with markedly different plume
heights, plume structure, and a factor of two increase in the magnitude of plume mixing ratios relative to the
lower grid cell size simulation, and these differences persist into the 24[th] simulation hour (Figure 5(e,f)). Mixing
ratio differences of these magnitudes are to be expected given the increase in resolution, but Figure 5 shows that
other important aspects of the predicted plumes have changed. The plume heights are a function of predicted
local stability conditions in the grid-square containing the source, and the variation shown here represents a
substantial change in the predicted local stability for the origin sources of these plumes, resulting from the change
in model horizontal grid cell size.

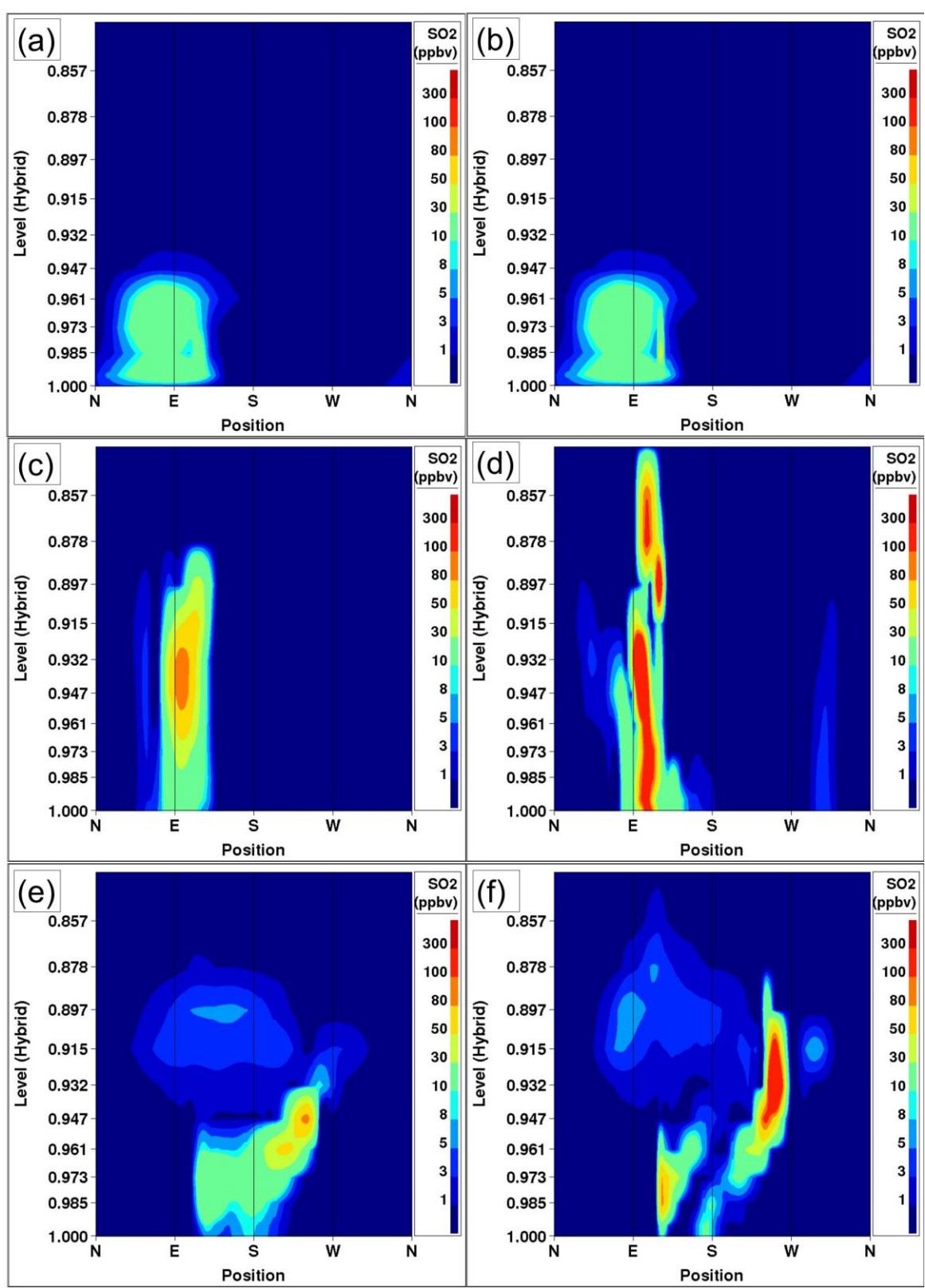


Figure 5.  Comparison of simulated SO$_2$ plume mixing ratios (ppbv) located 0.2km from a major point source, for OS2.5km

simulations (left column) and OS1km simulations (right column), at 0 (a,b), 12 (c,d), and 24 (e,f) hours into a 24 hour

simulation.


Figure 6 compares the maximum surface SO₂ during the entire period for each simulation, as well as the difference
in maximum SO₂ between the simulations, along with a scatterplot of OS2.5km versus OS1km simulation results.
In the latter two panels, OS2.5km values were assigned to the corresponding OS1km grid-cell locations using the
nearest-neighbour approach.

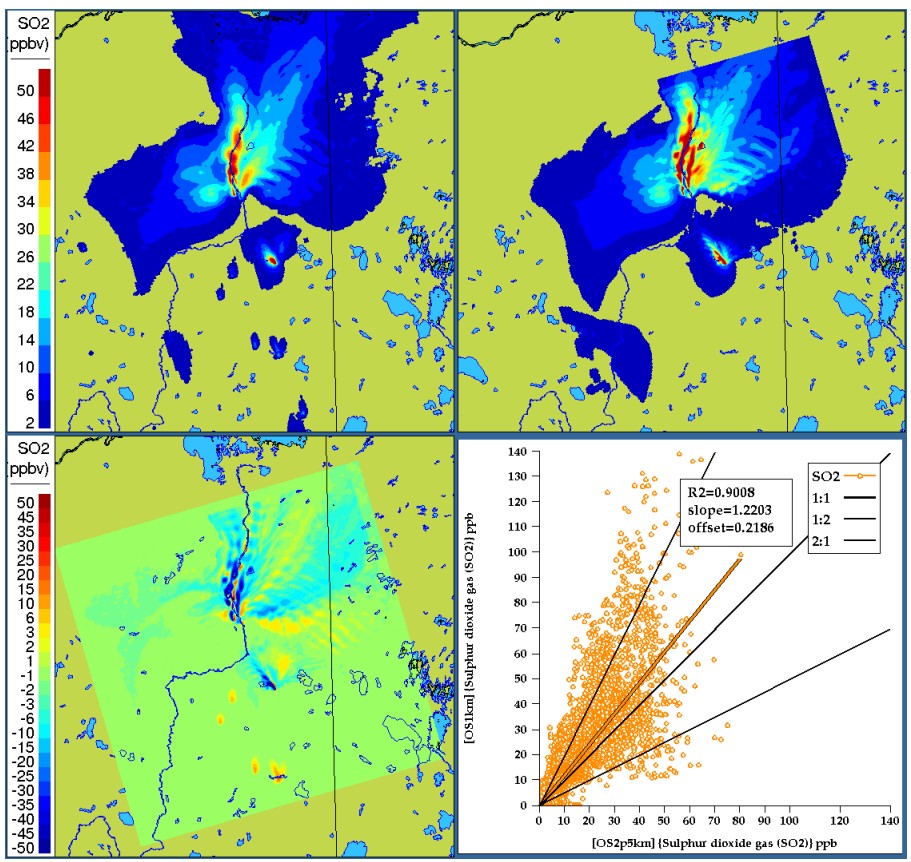

Figure 6.  Comparison of total-simulation *maximum* surface SO₂ mixing ratios (ppbv) at (a) 2.5km and (b) 1km grid cell size
(ppbv).   (c) Difference (2.5km – 1km).  (d) Scatterplot of 2.5km (x-axis) versus 1km (y-axis) total simulation average grid-cell
surface SO₂ mixing ratios.

The maximum surface concentrations tend to show more elongated structures at the smaller grid cell size,

comparing Figures 6(a,b), particularly for plumes in the western (left) half of the OS1km domain.  The difference

plot (Figure 6(c)) shows that local maximum concentration differences of up to -45 ppbv occur, due to changes in

the placement and maximum concentration of high concentration plumes.  The scatterplot of Figure 6(d) shows

that OS1km model has a demonstrated ability to achieve higher concentrations than the OS2.5km model, with a

slope of 1.22, and a noticeable clustering of values along the 1:2 line.  While these results are not unexpected

since approximately 95% of the SO₂ emissions in the domain originate in large stack, or point, sources, and hence

initial concentrations at source would be expected to 6.25x higher in the OS1km simulation, they also suggest that

a substantial improvement in the OS1km model's ability to capture SO₂ concentrations *should* be possible.  That is,

the results of the two models are substantially different, and given the reduction in numerical error expected with

employing a smaller grid cell size, the latter might be expected to outperform a larger grid cell size model.
However, as we shall demonstrate in the next section, plume placement errors such as depicted in Figure 3 play a
substantial role in model performance as grid cell size decreases.
3.2 Quantitative comparisons

3.2.1 Surface observation comparison
The locations of the local network of 10 surface monitoring stations located near the sources of emissions in the
region (oil sands facilities) are shown in Figure 7. As noted in section 2.4, we carry out several analyses:
(1) The standard evaluation (model values are extracted from the model grid-cells containing the observation
stations, at both grid cell sizes).
(2) Equal areas of representativeness, 1km and 2.5km grid cell sizes (the nearest nine OS1km grid cells are
compared to the OS2.5km single cell evaluation in two ways):
a. Averaging of the OS1km results across the nine grid cells prior to evaluation (to determine whether
the mean value is better represented by the smaller grid cell size, similar to the approach taken in
Kang *et al.* (2007)).
b. Selection of the *best* of the nine grid cells (closest to the observation value), to determine the extent
to which the OS1km model is capable of better representing the concentrations somewhere within
the corresponding OS2.5km model grid cell, if not at the OS1km cell closest to the observation
location. Higher scores for the 1km grid cell size simulation in this case would indicate that while
errors in plume positioning (for example due to errors in the synoptic scale flow) negate some of the
advantages of the OS1km simulation, the plume may be better represented by the OS1km simulation
within the 2.5km grid cell's area.
(3) Equal areas of representativeness and equal regions of variability (nearest nine 2.5km cells are compared to
the nearest forty-nine 1km cells). Here we make the assumption that the 2.5km grid cell size model's ability
to resolve features is limited to the surrounding three grid cells in each horizontal dimension, and make use of
the closest-in-size block of corresponding 1km cells (a $7 \times 7$ grid centered on the cell containing the
observation point). In both cases, the model value closest to the observations is chosen prior to evaluation.
While evaluations (2b) and (3) deliberately select the "best" value, they also provide a quantitative estimate of
the extent to which each model is capable of achieving the correct answer within roughly equal representative
areas centered on the observation station locations. These comparisons are intended to evaluate (a) the
extent to which the 1km grid cell size is capable of improving simulation results despite, *e.g.*, the larger scale
flow resulting in errors in the plume placement, and (b) whether the 1km grid cell size model is capable of
outperforming the 2.5km grid cell size model *over equivalent regions*. In the last test, we place both models on
an equal footing with regards to the region being represented, as well with regards to allowing cell-to-cell
variability and the selection of a closest match to observations.
Our evaluation is presented as tables of statistical metrics. The comparisons employing the nearest neighbour
approach are described with a "B#" superscript suffix, denoting that the "Best" sample within a square centred
on the observation point containing a total of # grid cells (*e.g.* the OS1km[B9] label denotes a comparison
between observed data and the simulation grid cell within a $3 \times 3$ grid-cell square centered about the
observation point). Similarly, an A# superscript describes a comparison between the observations and the
Average of the # square of grid cells centered on the observation point.
Comparisons to surface concentrations were performed using publicly available data collected by the Wood
Buffalo Environmental Association (WBEA), which operates the air-quality monitoring network residing within
the OS1km domain. The monitoring station locations are shown in Figure 7. The statistical performance of the
models, calculated using the procedure outlined above, are given in Tables 2 through 5, for $SO_2$, $NO_x$, $O_3$, and
$PM_{2.5}$, respectively.

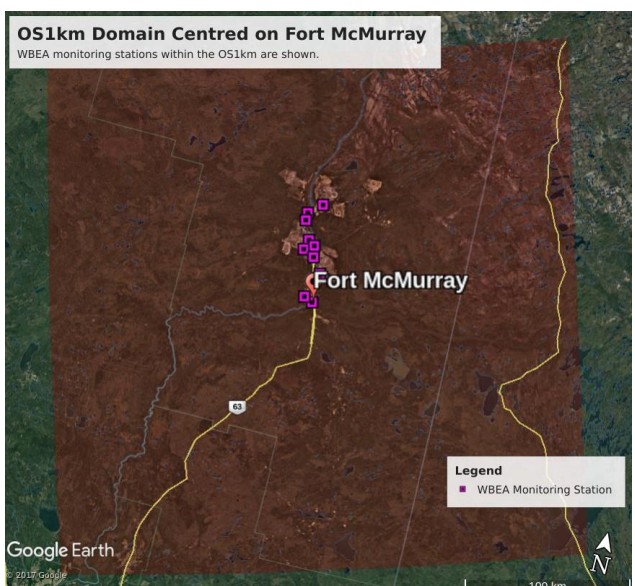
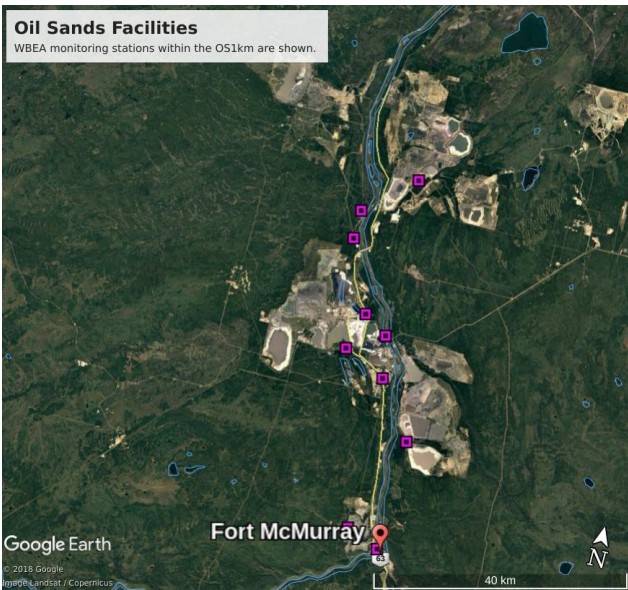


Figure 7. Illustration of the OS1km domain, with observation station locations. (a) Entire domain. (b) Close-up
view of station locations. Monitoring stations are shown as purple outline squares in both images. Light grey
regions in the background satellite image (b) are oil sands open-pit mining operations.
In the *standard* model grid cell to observation measurement comparison for $SO_2$, and $NO_x$ (first two columns,
Tables 2 and 3), the OS1km simulation had *worse* scores for all the metrics considered here. For $O_3$, the OS1km
model had the better score for the correlation coefficient and root mean square error, and worse scores for all
remaining model evaluation metrics. For $PM_{2.5}$, the OS1km model had higher performance for the correlation
coefficient and biases, while the OS2.5km model outperforms the OS1km model for all other metrics examined
here. Based on a standard analysis, the OS1km model thus performs poorly compared to the OS2.5km model; the
expected advantages associated with reduced numerical error in transport at smaller grid cell sizes are being offset
by other factors controlling the net model error.
When the standard evaluation is compared to the *average* of the nearest nine 1km simulation grid cells
surrounding the observation point (third column of the tables), an intermediate result appears. For $SO_2$ (Table 2)
the nine-cell OS1km average has the best performance for correlation coefficient - indicating a better time
distribution of events may be achieved by a nine cell average at 1km grid cell size. The other metrics for the A9
simulation are intermediate between the two standard evaluations for each simulation, indicating that some of the
performance loss resulting from the use of 1km grid cell size is reduced through averaging results to approximately
the same size regions as the OS2.5km grid cell size. The latter result holds for all metrics for $NO_x$ (including R, see
Table 3). For ozone (Table 4), averaging the nine nearest OS1km grid cells prior to measurement gives the best
performance for R and RMSE, and worse performance for the other metrics. For $PM_{2.5}$ (Table 5), all metrics for the
OS1km nine grid-cell average aside from the bias fall mid-way between the two standard methodology evaluations.
Averaging the smaller grid cell size model results thus shows a marginal improvement, depending on the species,
but overall does not compensate for the decrease in performance resulting from going to the smaller grid cell size.
We next ask the question, "Does a more accurate simulation value *exist* within the same region of the 1km model
as is encompassed by a 2.5km grid cell?" (fourth column of these Tables), by selecting the model value in the
nearest nine 1km grid cells with the closest match to observations and comparing to the corresponding single
2.5km grid cell. A dramatic improvement in the relative OS1km performance metric scores can be seen. For each
of Tables 2 through 5, this "best of nine" 1km comparison outperforms the previous 3 comparisons (columns 1
through 3), for all metrics. These improvements are sometimes dramatic (*e.g.* a doubling of correlation coefficient
along with a reduction in mean bias by a factor of three, a reduction of $NO_x$ mean bias values by a factor of 3, a shift
of coefficient of error from negative to positive values for $O_3$, and a reduction in the coefficient of error for $PM_{2.5}$ by
a factor of 2.5 compared to the nearest competing value from the previous evaluations. The coefficient of
efficiency for $SO_2$ and $O_3$ make the transition from negative to positive values when the "best-of-nine" methodology
is used, indicating that the model is able to better predict the observations than the observed mean, somewhere
within an equivalent area. This evaluation suggests that the OS1km model does *contain* a better result within the
same approximate region encompassed by a 2.5km grid cell. However, the location of that better result may be
subject to positioning error, such as described in Figure 3.
A valid argument could be made that the methodology employed in this fourth evaluation is subject to selection
bias, in that the selection of a *best* value in the case of the nearest nine 1km simulation places that model
simulation at an advantage relative to the 2.5km model. To address this last issue, the final two additional
methodologies for evaluation were employed, still maintaining the same approximate area of representativeness
for a grid cell, namely choosing the best value out of the nearest *nine* 2.5km grid cells (the limiting resolution of this
model simulation), and the best value out of the nearest *forty-nine* 1km grid cells (fifth and sixth columns of Tables
2 through 5, respectively).  That is, we attempt to place the two models on an equal basis with regards to selection
bias within a given region containing an observation station.
Two important results can be seen from this final evaluation.  First, as was the case for the "Best of 9" for the
OS1km simulation compared to the standard OS1km evaluation, the "Best of 9" for the OS2.5km simulation has a
considerably better performance than the standard OS2.5km evaluation (compare fifth and first columns, Tables 2
through 5). That is, the OS2.5km model may *also* be subject to location errors in transported species representation
which influence model performance.  However, when performance within the 56.25 $km^2$ area surrounding each
measurement point in the OS2.5km "Best of 9" evaluation is compared to the 49 $km^2$ area surrounding the
measurement points in the OS1km "Best of 49" simulation (*i.e.* compare columns five and six in Tables 2 through 5),
it can be seen that the OS1km model outperforms the OS2.5km model for all metrics for $O_3$, and $PM_{2.5}$, and all
metrics aside from bias for $SO_2$ and $NO_x$.   That is, despite the OS1km model having a slight disadvantage in the
relative size of the representative area containing the measurement station location, and both models being
allowed a similar selection strategy, the OS1km model is capable of generating values closer to the observations
than the OS2.5km model within an equivalent sub-region, across most of the metrics and chemical species
considered here.
This final result is strongly suggestive of the presence of issues such as illustrated in Figure 3.  These may include
errors in the larger scale synoptic wind flow, combined with the reduced size of plumes as grid cell size is reduced,
leading to more "misses" than "hits" for a given recorded event at a measurement station compared to the coarse
grid cell size model.  There may be multiple additional causes for such errors (examples include poor observation
density in the region for model initialization, underlying lower resolution boundary condition fields such as
topography not improving with the reduction in grid cell size, inaccuracies in land use fields used in meteorological
modelling due to rapid development, and errors in other aspects of the reaction transport modelling system aside
from horizontal resolution).  The expected advantages of the small grid cell size, such as better representation of
the concentrations of species within plumes and hence better representation of their reactive chemistry (c.f.
Lonsdale *et al.,* 2012), may be lost in a standard performance analysis due to these other issues.
Our analysis suggests that a practical limit in the benefits of increasing model accuracy may be reached when
resolution exceeds some threshold, as a result of other errors inherent in the modelling system.  However, the
analysis also suggests that if these non-resolution-related errors are corrected, the benefits of adopting a smaller
grid cell size may be substantial.  For example, meteorological data assimilation employing a dense monitoring
network for a specific area of interest would be expected to show a greater impact for smaller than larger grid cell
sizes, due to the greater ability of the former to take advantage of the observation density in correcting the initial
meteorological state. We note that recent work applying land use data assimilation (Carrera *et al.*, 2015) to
regional 2.5km grid cell size weather simulations (Milbrandt *et al.,* 2016) have suggested that such data assimilation
may indeed improve forecast skill at the very local scale.
Table 2. Surface $SO_2$ observations to model comparison for entire simulation period (ppbv)

| Evaluation Metric | OS2.5km | OS1km | OS1km[A9] | OS1km[B9] | OS2.5km[B9] | OS1km[B49] |
|---|---|---|---|---|---|---|
| Index of Agreement | 0.237 | 0.154 | 0.207 | 0.601 | 0.701 | 0.810 |
| Pearson Correlation Coefficient | 0.290 | 0.230 | 0.295 | 0.604 | 0.672 | 0.848 |
| Normalized Mean Gross Error | 2.128 | 2.363 | 2.212 | 1.114 | 0.834 | 0.529 |
| Mean Gross Error | 2.918 | 3.240 | 3.034 | 1.528 | 1.143 | 0.725 |
| Coefficient of Error | -0.525 | -0.693 | -0.585 | 0.202 | 0.403 | 0.621 |
| Root Mean Square Error | 7.063 | 9.665 | 7.876 | 4.436 | 3.671 | 2.618 |
| Normalized Mean Bias | 1.130 | 1.376 | 1.299 | 0.347 | -0.010 | 0.017 |
| Mean Bias | 1.550 | 1.887 | 1.781 | 0.475 | -0.013 | 0.024 |

• 5466 Samples used
Table 3. Surface $NO_x$ observations to model comparison for entire simulation period (ppbv)

| Evaluation Metric | OS2.5km | OS1km | OS1km[A9] | OS1km[B9] | OS2.5km[B9] | OS1km[B49] |
|---|---|---|---|---|---|---|
| Index of Agreement | 0.177 | 0.138 | 0.152 | 0.416 | 0.589 | 0.665 |
| Pearson Correlation Coefficient | 0.143 | 0.114 | 0.116 | 0.165 | 0.305 | 0.388 |
| Normalized Mean Gross Error | 1.520 | 1.593 | 1.567 | 1.079 | 0.760 | 0.619 |
| Mean Gross Error | 12.898 | 13.518 | 13.296 | 9.156 | 6.447 | 5.255 |
| Coefficient of Error | -0.646 | -0.725 | -0.697 | -0.168 | 0.177 | 0.329 |
| Root Mean Square Error | 28.052 | 35.197 | 34.644 | 25.782 | 15.315 | 13.704 |
| Normalized Mean Bias | 0.493 | 0.570 | 0.542 | 0.174 | -0.027 | -0.063 |

| | | | | | | |
|---|---|---|---|---|---|---|
| Mean Bias | 4.183 | 4.834 | 4.597 | 1.477 | -0.231 | -0.531 |

- 3257 Samples used

Table 4. Surface $O_3$ observations to model comparison for entire simulation period (ppbv)

| Evaluation Metric | OS2.5km | OS1km | OS1km[A9] | OS1km[B9] | OS2.5km[B9] | OS1km[B49] |
|---|---|---|---|---|---|---|
| Index of Agreement | 0.414 | 0.405 | 0.404 | 0.527 | 0.637 | 0.690 |
| Pearson Correlation Coefficient | 0.496 | 0.506 | 0.515 | 0.606 | 0.688 | 0.738 |
| Normalized Mean Gross Error | 0.660 | 0.670 | 0.672 | 0.534 | 0.410 | 0.349 |
| Mean Gross Error | 10.757 | 10.915 | 10.949 | 8.692 | 6.673 | 5.687 |
| Coefficient of Error | -0.172 | -0.189 | -0.193 | 0.053 | 0.273 | 0.380 |
| Root Mean Square Error | 16.040 | 15.859 | 15.794 | 13.305 | 11.084 | 9.719 |
| Normalized Mean Bias | 0.527 | 0.559 | 0.579 | 0.463 | 0.337 | 0.304 |
| Mean Bias | 8.579 | 9.104 | 9.431 | 7.536 | 5.488 | 4.945 |

- 2189 Samples used

Table 5. Surface $PM_{2.5}$ observations to model comparison for entire simulation period ($\mu$g m$^{-3}$)

| Evaluation Metric | OS2.5km | OS1km | OS1km[A9] | OS1km[B9] | OS2.5km[B9] | OS1km[B49] |
|---|---|---|---|---|---|---|
| Index of Agreement | 0.280 | 0.262 | 0.267 | 0.412 | 0.508 | 0.572 |
| Pearson Correlation Coefficient | 0.201 | 0.216 | 0.214 | 0.314 | 0.376 | 0.466 |
| Normalized Mean Gross Error | 0.791 | 0.811 | 0.806 | 0.647 | 0.541 | 0.471 |
| Mean Gross Error | 5.342 | 5.478 | 5.441 | 4.365 | 3.651 | 3.181 |
| Coefficient of Error | -0.439 | -0.476 | -0.466 | -0.176 | 0.016 | 0.143 |
| Root Mean Square Error | 8.286 | 8.786 | 8.663 | 7.117 | 6.169 | 5.690 |
| Normalized Mean Bias | -0.268 | -0.257 | -0.257 | -0.289 | -0.299 | -0.287 |
| Mean Bias | -1.812 | -1.734 | -1.736 | -1.948 | -2.016 | -1.937 |

• 3377 Samples used

The surface observation data were also analyzed by time-of-day, with both observations and simulations split into
daytime (hours 9:00 to 18:00 local time) and nighttime (hour 19:00 to 8:00 local time) data pairs (Appendix, Tables
A1 through A8, Carslaw and Ropkins, 2012).  Within each of these diurnally segregated time periods, the broad
aspects of the comparison were the same as for the "all data" Tables 2 to 5 above: the OS1km simulations tendied
to have reduced performance in a standard analysis, averaging improved but not completely ameliorated the
performance of the OS1km simulation, a methodology employing the best of nine OS1km grid cells had superior
performance to the two standard comparisons, and comparison of the "best of" methodologies for equal areas
showed better performance for the OS1km compared to the OS2.5km simulation.  We also noted substantial
differences in the day and night performance of both models across the methodologies.  For example, daytime $SO_2$
and $NO_x$ performance within a given model and comparison methodology was usually better than nighttime
performance for IOA,R, NMGE, COE and NMB, while worse for RMSE, while nighttime $O_3$ performance was better
for IOA, r, NMGE, and COE.  Daytime $PM_{2.5}$ performance was better than nighttime for IOA, r, COE, and NMB.  The
study area is located in a broad river valley with frequent slope-defined anabatic/akatabic and drainage flow
events.  These often have a diurnal nature, and may explain part of the day/night differences.  Example sources of
these differences may include the relative ability of the driving meteorological model to capture daytime versus
nighttime mixed layer turbulence and the planetary boundary layer height.
3.2.2 Comparisons to Aircraft Observations
Twenty-two aircraft observation flights were carried out during the study simulation period – we present
statistical comparisons using the standard approach only, here (model grid cell containing the observation point to
observation data at the aircraft location).  Model values were linearly interpolated in time and space to the
aircraft observation locations and times (aircraft observations were on a 10s interval.)  We begin with a composite
comparison across all observation times, in Table 6.
Table 6. Aircraft observation comparisons, $SO_2$ and $NO_2$ (ppbv)

|  | $SO_2$ (21787 samples) | | $NO_2$ (18310 samples) | |
| --- | --- | --- | --- | --- |
|  | OS2.5km | OS1km | OS2.5km | OS1km |
| Index of Agreement | 0.63 | 0.62 | 0.61 | 0.58 |
| Pearson Correlation Coefficient | 0.26 | 0.28 | 0.39 | 0.34 |
| Normalized Mean Gross Error | 1.07 | 1.09 | 0.90 | 0.96 |
| Mean Gross Error | 3.98 | 4.06 | 1.56 | 1.68 |
| Coefficient of Error | 0.27 | 0.25 | 0.23 | 0.17 |
| Root Mean Square Error | 12.84 | 13.97 | 3.12 | 3.62 |
| Normalized Mean Bias | -0.31 | -0.29 | -0.26 | -0.20 |
| Mean Bias | -1.17 | -1.07 | -0.45 | -0.34 |


The results are in general similar to the surface analysis, in that the OS1km simulation tended to have worse performance than the OS2.5km simulation (exceptions being the biases for both $SO_2$ and $NO_2$, and the slightly better OS1km correlation coefficient for $SO_2$). One striking difference between the first two columns of Tables 2 and 3 and Table 14 are the magnitude of the differences between the simulations. Aloft (Table 6), the differences in performance metric magnitudes between OS2.5km and OS1km simulations are much smaller than at the surface (Tables 3 and 4). The biases are negative aloft, while positive at the surface, indicating that both models may be lofting plumes to insufficient distances; one of the possible (non-horizontal grid cell size dependent) causes of model error may be in the extent of vertical transport. This possibility is examined in more detail in Akingunola *et al.* (2018, and Gordon *et al.* (2018). An example of this behaviour is shown in Figure 8; both plumes fumigate to the surface, while the observed plume resides largely aloft. The OS1km model captures the higher concentrations to a better degree, but the impact of excessive fumigation more than offsets this improvement, as is shown by the performance evaluation of Table 7, where both models have negative biases aloft. In this particular case, the tendency of the model to overestimate the extent of fumigation has a bigger impact on performance than grid cell size. Garcia-Menendez *et al.* (2014) have noted similar results for forest fire plume prediction.

Panels (a) and (c) of Figure 8 provide a further example of the kind of situation referenced in Figure 3; surface monitoring station locations are depicted as grey circles, one of which is identified with a pink arrow. This station lies within the plume at 2.5km resolution (Figure 8(a)), and outside of the plume at 1km resolution (Figure 8(c)). While the plume direction is the same at both scales, that is, the large-scale wind field controls the positioning of the plume axis, the smaller grid cell size simulation places a stronger constraint on the accuracy of the wind field. For example, if the simulated large-scale flow direction was inaccurately predicted by only a few degrees, the plume would not appear in the 1km simulation time series at this location, while registering as present in the 2.5km simulation. Nevertheless, the plume maximum concentration is better captured by the smaller grid cell size simulation (compare maximum values in observed aircraft $SO_2$, Figure 8 (b, d)). The higher resolution simulation may thus more accurately simulate the plume maximum concentration – but not its placement in space, as was hypothesized in Figure 3.

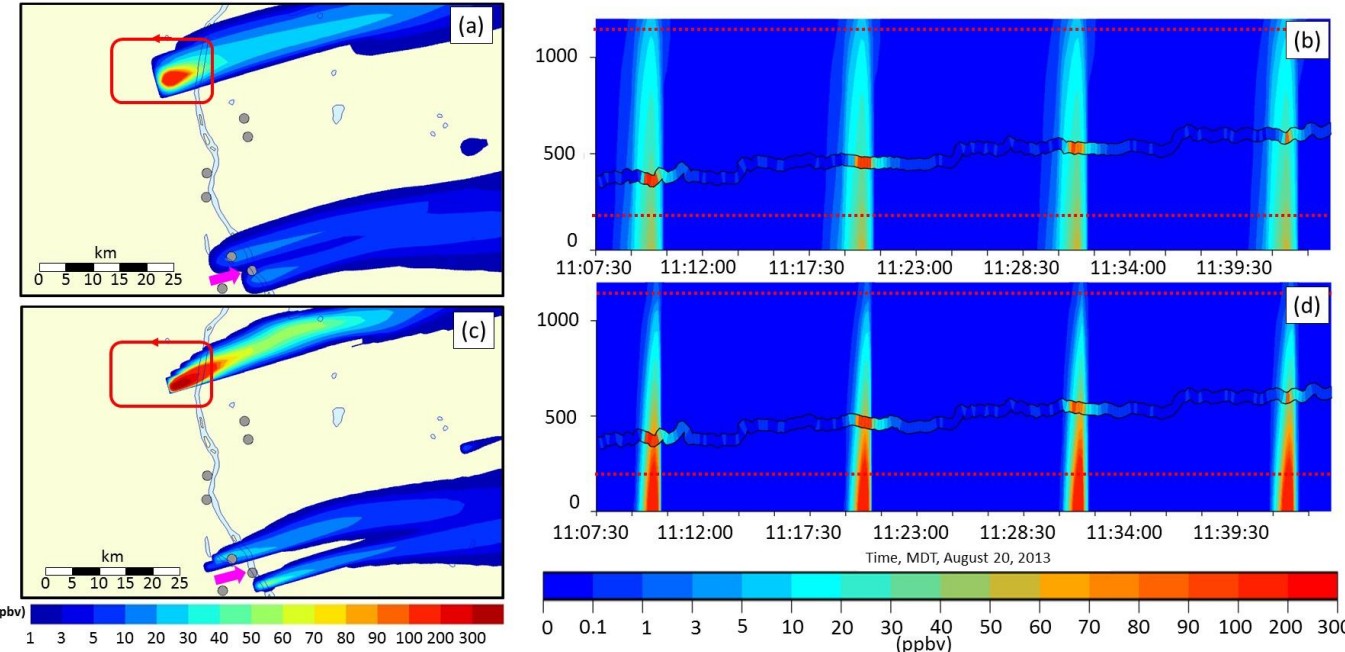

Figure 8. Comparison between OS2.5km (a,b) and OS1km (c,d) simulations for $SO_2$ relative to aircraft observations (ppbv). (a,c): Simulated surface concentrations of $SO_2$, with the flight track shown as a red line. Grey circles: surface monitoring station locations; pink arrow indicates a station located inside a plume at 2.5km resolution (a), and outside the plume at 1km resolution (c). (b,d): Portion of the simulated concentration profiles along the flight path as a function of time. Successive intersections of the flight path with the plume appear as background colour contours; observed $SO_2$ aboard the aircraft is shown between the two black lines. Vertical axis is elevation above the ground; the aircraft elevation is increasing with successive passes around the facility. Dotted lines show the upper and lower vertical extent of the observed plume; note that for both model simulations, the plume erroneously fumigates the surface.

Table 7.  Standard performance evaluation of Flight 8 for $SO_2$ (ppbv)

|  | OS2.5km | OS1km |
|---|---|---|
| Index of Agreement | 0.69 | 0.68 |
| Pearson Correlation Coefficient | 0.42 | 0.31 |
| Normalized Mean Gross Error | 1.04 | 1.09 |
| Mean Gross Error | 4.02 | 4.25 |
| Coefficient of Error | 0.39 | 0.35 |
| Root Mean Square Error | 16.72 | 20.57 |
| Normalized Mean Bias | -0.42 | -0.34 |
| Mean Bias | -1.63 | -1.32 |

1261 samples used.

Meanwhile other flights show a clear advantage of the OS1km model.  One example is given by the $NO_2$
performance evaluation of Table 8 and depicted in Figure 9, for Flight 17 (a similar flight plan carried out around
the same facility as Flight 8).  While the correlation coefficient degraded slightly in the OS1km resolution
simulation, all other performance measures were improved with the decrease in grid cell size.  Two time versus
height profile cross-sections for Flight 17 are shown in Figure 9.  In the upper two panels, the OS2.5km (Figure
9(a)) and OS1km (Figure 9(b)) simulations are compared for the portion of the overall flight track circling the given
facility.  This comparison clearly shows that the OS1km model does a better job of capturing the width of the high
concentration region of the plume – however, the location of the model plume lags the observations.  During this
portion of the flight alone, the OS2.5km model statistics, particularly the correlation coefficient, outperform the
OS1km model, due to this issue of plume location mismatching.  Figures 9(a,b) may be compared to Figure 3(a,b) –
the same situation is depicted in both Figures.  Figure 9(c,d) show the OS2.5km simulation (10(c)) and OS1km
simulation results in another portion of the flight – here the OS1km performance for most statistics was better
than the OS2.5km model performance.  The OS1km model (Figure 9(d)) captures the existence of a lower
concentration layer aloft in the right-hand side of the cross-section, and the existence of low concentration
intervening layers, as well as the overall lower concentrations of $SO_2$, while the OS2.5km model does not resolve
these fine scale and lower concentration features.  We note here that IoA, CoE and the other error measures
capture the visual impression that the OS1km model outperforms the OS2.5km model for this flight, while the
correlation coefficient is highly dependent on the placement of the plume maximum in the upper two panels.
These and the snap-shot comparisons described in Section 3.1 show that the higher resolution model is having a
significant impact on predictions – however, other aspects of the overall model performance are preventing the
potential benefits of higher resolution from influencing the standard performance evaluation.


Table 8. Standard performance evaluation of Flight 17 for $NO_2$ (ppbv)

|  | OS2.5km | OS1km |
|---|---|---|
| Index of Agreement | 0.26 | 0.58 |
| Pearson Correlation Coefficient | 0.26 | 0.25 |
| Normalized Mean Gross Error | 2.03 | 1.15 |
| Mean Gross Error | 0.52 | 0.29 |
| Coefficient of Error | -0.48 | 0.16 |
| Root Mean Square Error | 1.37 | 0.70 |
| Normalized Mean Bias | 0.83 | -0.54 |
| Mean Bias | 0.21 | -0.14 |

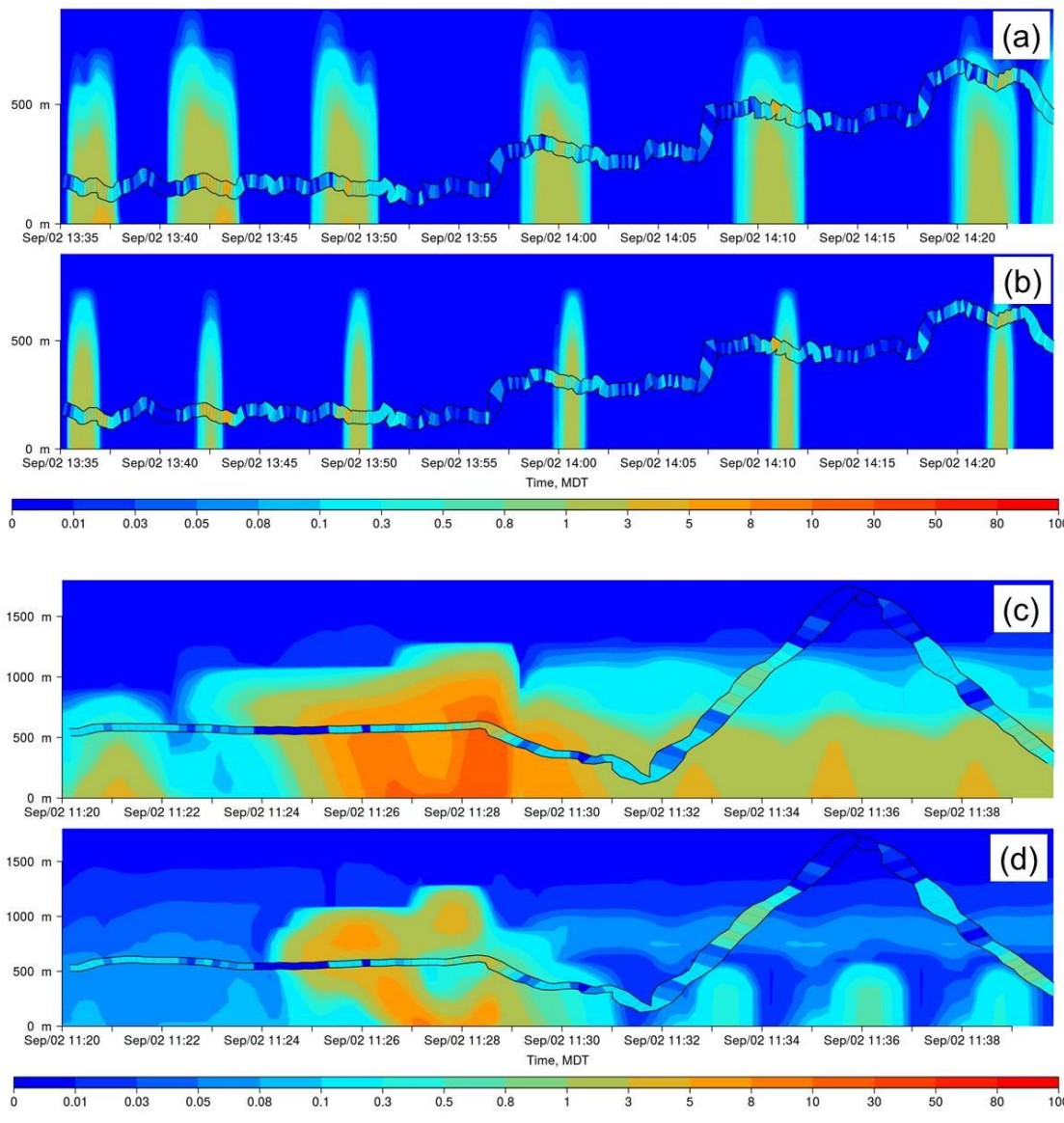


Figure 9. Flight 17 comparison for $NO_2$ (ppbv) for portions of the net flight track circling the CNRL facility for
OS2.5km (a) and OS1km (b) simulations, and for a later section of the same flight path for the OS2.5km (c) and
OS1km (d) simulations.

27

## 4. Discussion

A key result of our current work is that 1km grid cell size simulations resulted in improved prediction of plume concentration maxima relative to 2.5km grid cell size simulations, despite having no improvement using standard scoring methodologies. We also have described a scoring approach wherein these potential advantages of higher resolution may be quantified. We believe that flow field effects such as described in Figure 3 are a general result of increasing grid resolution, but note important caveats, which include:

(1) The availability of meteorological observation and high resolution emissions data to provide model driving information, and the resolution and proximity of this information to the simulation location. Both will influence the relative importance of grid cell size on model results. If this information is available in a higher resolution than the lower of two grid cell size simulations being compared, and/or is used via data assimilation to improve model initial meteorological conditions, our expectation is that the smaller grid cell size model may outscore the larger grid cell size model, even for more standard metrics.

(2) The extent to which local, versus synoptic, weather conditions drive flow in a given region. For example, in the urban heat island meteorological simulations of Leroyer *et al.* (2014), the accuracy of local flow predictions was shown to be extremely dependent on the representation of the urban heat island, and the accuracy of the latter was critically dependent on the grid cell size (which in this example went down to 250 m). In this respect, for meteorological conditions wherein local factors can dominate the flow, and where those conditions may be adequately modelled only at very high resolution, we would again expect the smaller grid cell size simulation to provide better performance, for standard metrics.

(3) Conversely, model performance using standard metrics should not be expected to *increase* with successively larger and larger grid sizes; the accuracy of even the synoptic flow field will not be captured as model resolution decreases.

Given these considerations, we recommend that modellers should attempt successively smaller grid cell sizes to determine the following: first, the point at which, for their particular system and simulation location, subsequent grid cell size reductions fail to improve performance; and second, to make use of still higher resolutions for studies wherein the point-to-point comparison is less important, and other factors such as accurately capturing the plume chemistry are more crucial.

## 5. Summary and Conclusions

Our work suggests the following:

Decreasing air-quality model horizontal grid cell size will not necessarily result in improvements to model performance in standard performance evaluations, in which the model values at the grid-cells encompassing measurement location stations are used in a pairwise comparison to observations. Other considerations, such as

the accuracy of the larger scale wind direction and speed forecast, and the accuracy of the plume rise
parameterization used within the model may play a greater role in the overall performance of the model, and
reduce the benefits of the smaller grid cell size. In the context of a standard model performance evaluation, there
may be fixed limits to the benefits of decreasing model grid cell size.
Despite this difficulty, our results also show that the use of smaller grid cell sizes have some potential benefits, in
that these models do a better job of resolving specific air pollution features, like high concentration maxima
within plumes. Both coarse and fine grid cell size plumes may be misplaced in both time and space, with the net
result that the latter model has a worse performance in a standard comparison, but is nevertheless more likely to
capture the correct in-plume concentrations, and hence the chemistry, of the actual plume, in the *neighbourhood*
of the observation location. When the evaluation is broadened to find the closest fit to observations in the vicinity
of the observation station, with models confined to a similar representative area around the observation station,
these potential benefits of the smaller grid cell size become apparent.
Our results should not be taken as an indication that the standard metrics for model comparison are in some way
flawed – they provide the most rigorous method for evaluating the performance of a model at specific monitoring
locations and specific times. However, the ancillary performance assessment methodology presented here shows
that models with very small grid sizes, which may have standard performance metric scores that have not
improved or even have degraded relative to larger grid cell size models, nevertheless have scientific value, in
terms of being better able to capture plume concentrations and hence plume chemistry, if not plume position.
The work also suggests that the prediction accuracy of very local transport conditions may be a large factor in
preventing the smaller grid cell size models from achieving improved performance in standard performance
analyses.
These findings suggest that at the current state of development, VHR air-quality models are of benefit for the
specific purpose of chemical process studies, in which the main aim of the work is to accurately simulate plume
chemistry – and in which accurate forecasting of the *position* of the plume in time and space is a secondary
concern. Our work also suggests that efforts to improve other aspects of the overall modelling framework which
improve the large-scale flow (for example, the use of data assimilation of local meteorology to improve wind
direction predictions) may result in greater benefits as smaller grid cell sizes are employed.

*Author contribution:* M.R.: computer simulations and analysis, graphical outputs, initial manuscript draft; A.H.: supervision of
M.R., research advice and infrastructure, manuscript writing, comments on manuscript drafts. P.A.M.: co-supervision of M.R.,
research advice and infrastructure, manuscript writing, lead for revisions and responses to referees. A.A.: model code
assistance and setup, provision of model – observation comparison and scoring package. J.Z.: provision of 2.5km and 1km
resolution emissions files. M.D.: provision of 2.5km and 1km resolution emissions files, comments on manuscript drafts. Q.Z.:
provision of 2.5km and 1km resolution emissions files.
*Acknowledgements.* The authors wish to thank the support of Environment and Climate Change Canada
(ECCC), under the CCAP program, for supporting this research.  The authors also gratefully acknowledge the
assistance of Michel Valin and Sylvie Gravel for advice and assistance with the installation of GEM-MACH on the
Carleton University workstations during the early stages of this project.

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

# 7. Appendix A: Model Evaluation Statistics

Table A1: Model Comparison Statistics

| Metric and Formula | Range | Ideal Score |
|---|---|---|
| $Index\ of\ Agreement (IOA)$ $$= \begin{cases} 1 - \dfrac{\sum |M_i - O_i|}{2(O_i - \bar{O})}, when \sum |M_i - O_i| \le 2(O_i - \bar{O}) \\ \dfrac{2(O_i - \bar{O})}{\sum |M_i - O_i|} - 1, when \sum |M_i - O_i| > 2(O_i - \bar{O}) \end{cases}$$ | [-1,1] | 1 |
| $Coefficient\ of\ Error\ (COE) = 1 - \dfrac{\sum |M_i - O_i|}{(O_i - \bar{O})}$ | [-∞, 1] | 1 |
| $Mean\ Bias\ (MB) = \dfrac{1}{N} \sum (M_i - O_i) = \bar{M} - \bar{O}$ | | 0 |
| $Mean\ Gross\ Error\ (MGE) = \dfrac{1}{N} \sum |M_i - O_i|$ | | 0 |
| $Normalized\ Mean\ Bias\ (NMB) = \dfrac{\sum (M_i - O_i)}{\sum O_i} = \left( \dfrac{\bar{M}}{\bar{O}} - 1 \right)$ | | 0 |
| $Normalized\ Mean\ Gross\ Error\ (NMGE) = \dfrac{\sum |M_i - O_i|}{\sum O_i}$ | | 0 |
| $Root\ Mean\ Square\ Error\ (RMSE) = \sqrt{\dfrac{1}{N} \sum (M_i - O_i)^2}$ | | 0 |
| $Pearson\ Correlation\ Coefficient\ (r) = \dfrac{\sum (M_i - \bar{M})(O_i - \bar{O})}{\sqrt{\sum (M_i - \bar{M})^2 \sum (O_i - \bar{O})^2}}$ | [-1.1] | 1 |

The limits on the summations were removed for brevity; all are from i = 1 to N where N is the number of observation-model pairs, $M_i$ is the i'th model value, O is the i'th observation value, and $\bar{M}, \bar{O}$ are the model and observed mean values, respectively.

## 7. Appendix B: Day Versus Night model performance for the different testing methodologies

Table B1. Surface $SO_2$ observations to model comparison, daytime (9:00-18:00) (ppbv).

|       | OS2.5km | OS1km  | OS1km[A9] | OS1km[B9] | OS2.5km[B9] | OS1km[B49] |
|-------|---------|--------|-----------|-----------|-------------|------------|
| IoA   | 0.374   | 0.286  | 0.352     | 0.712     | 0.762       | 0.872      |
| r     | 0.295   | 0.215  | 0.307     | 0.701     | 0.742       | 0.903      |
| NMGE  | 1.739   | 1.982  | 1.798     | 0.799     | 0.660       | 0.356      |
| MGE   | 4.201   | 4.788  | 4.343     | 1.931     | 1.595       | 0.860      |
| CoE   | -0.253  | -0.428 | -0.295    | 0.424     | 0.524       | 0.744      |
| RMSE  | 9.317   | 13.388 | 10.275    | 5.171     | 4.652       | 2.996      |
| NMB   | 0.730   | 0.990  | 0.871     | 0.054     | -0.166      | -0.118     |
| MB    | 1.764   | 2.391  | 2.104     | 0.132     | -0.401      | -0.286     |

- 2119 Samples used

Table B2. Surface $SO_2$ observations to model comparison, nighttime (18:00-9:00) (ppbv).

|       | OS2.5km | OS1km  | OS1km[A9] | OS1km[B9] | OS2.5km[B9] | OS1km[B49] |
|-------|---------|--------|-----------|-----------|-------------|------------|
| IoA   | -0.215  | -0.248 | -0.233    | 0.231     | 0.473       | 0.609      |
| r     | 0.204   | 0.206  | 0.205     | 0.339     | 0.421       | 0.620      |
| NMGE  | 3.143   | 3.281  | 3.215     | 1.896     | 1.300       | 0.964      |
| MGE   | 2.061   | 2.152  | 2.108     | 1.243     | 0.852       | 0.632      |
| CoE   | -1.549  | -1.607 | -1.607    | -0.537    | -0.054      | 0.218      |
| RMSE  | 5.055   | 5.450  | 5.450     | 3.802     | 2.858       | 2.313      |
| NMB   | 2.166   | 2.328  | 2.328     | 1.076     | 0.394       | 0.361      |
| MB    | 1.421   | 1.527  | 1.527     | 0.706     | 0.258       | 0.230      |

- 3347 Samples used

Table B3. Surface $NO_x$ observations to model comparison, daytime (9:00-18:00) (ppbv).

|       | OS2.5km | OS1km  | OS1km[A9] | OS1km[B9] | OS2.5km[B9] | OS1km[B49] |
|-------|---------|--------|-----------|-----------|-------------|------------|
| IoA   | 0.485   | 0.440  | 0.465     | 0.639     | 0.712       | 0.789      |
| r     | 0.254   | 0.259  | 0.270     | 0.427     | 0.507       | 0.680      |
| NMGE  | 0.927   | 1.009  | 0.962     | 0.650     | 0.519       | 0.380      |
| MGE   | 7.502   | 8.160  | 7.786     | 5.259     | 4.198       | 3.077      |
| CoE   | -0.030  | -0.120 | -0.069    | 0.278     | 0.424       | 0.577      |
| RMSE  | 14.843  | 15.811 | 15.571    | 11.272    | 9.982       | 7.964      |
| NMB   | -0.205  | -0.069 | -0.135    | -0.258    | -0.258      | -0.216     |
| MB    | -1.659  | -0.559 | -1.091    | -2.089    | -2.091      | -1.744     |

- 1252 Samples used

Table B4. Surface NO$_x$ observations to model comparison, nighttime (18:00-9:00) (ppbv).

|  | OS2.5km | OS1km | OS1km$^{A9}$ | OS1km$^{B9}$ | OS2.5km$^{B9}$ | OS1km$^{B49}$ |
|---|---|---|---|---|---|---|
| IoA | -0.016 | -0.050 | -0.045 | 0.275 | 0.511 | 0.587 |
| R | 0.113 | 0.081 | 0.083 | 0.118 | 0.240 | 0.295 |
| NMGE | 1.913 | 1.982 | 1.971 | 1.366 | 0.920 | 0.777 |
| MGE | 17.235 | 17.858 | 17.756 | 12.306 | 8.291 | 7.004 |
| CoE | -1.032 | -1.105 | -1.093 | -0.451 | 0.023 | 0.174 |
| RMSE | 35.003 | 44.669 | 43.972 | 32.797 | 18.475 | 16.875 |
| NMB | 0.958 | 0.988 | 0.990 | 0.458 | 0.126 | 0.039 |
| MB | 8.634 | 8.899 | 8.915 | 4.124 | 1.139 | 0.350 |

- 1862 Samples used

Table B5. Surface O$_3$ observations to model comparison, daytime (9:00-18:00) (ppbv).

|  | OS2.5km | OS1km | OS1km$^{A9}$ | OS1km$^{B9}$ | OS2.5km$^{B9}$ | OS1km$^{B49}$ |
|---|---|---|---|---|---|---|
| IoA | 0.141 | 0.192 | 0.184 | 0.338 | 0.396 | 0.529 |
| r | 0.166 | 0.215 | 0.211 | 0.327 | 0.367 | 0.504 |
| NMGE | 0.660 | 0.621 | 0.627 | 0.508 | 0.464 | 0.361 |
| MGE | 14.427 | 13.568 | 13.703 | 11.111 | 10.143 | 7.901 |
| CoE | -0.718 | -0.616 | -0.632 | -0.323 | -0.208 | 0.059 |
| RMSE | 21.209 | 20.063 | 20.035 | 16.714 | 15.140 | 12.466 |
| NMB | 0.587 | 0.542 | 0.557 | 0.454 | 0.414 | 0.326 |
| MB | 12.839 | 11.854 | 12.187 | 9.918 | 9.050 | 7.121 |

- 864 Samples used

Table B6. Surface O$_3$ observations to model comparison, nighttime (18:00 to 9:00) (ppbv).

|  | OS2.5km | OS1km | OS1km$^{A9}$ | OS1km$^{B9}$ | OS2.5km$^{B9}$ | OS1km$^{B49}$ |
|---|---|---|---|---|---|---|
| IoA | 0.451 | 0.398 | 0.399 | 0.534 | 0.719 | 0.727 |
| r | 0.526 | 0.541 | 0.557 | 0.642 | 0.784 | 0.784 |
| NMGE | 0.706 | 0.775 | 0.773 | 0.600 | 0.361 | 0.352 |
| MGE | 8.326 | 9.132 | 9.116 | 7.070 | 4.258 | 4.145 |
| CoE | -0.097 | -0.203 | -0.201 | 0.068 | 0.439 | 0.454 |
| RMSE | 11.236 | 12.029 | 11.974 | 10.297 | 6.935 | 7.137 |
| NMB | 0.492 | 0.624 | 0.651 | 0.510 | 0.262 | 0.296 |
| MB | 5.799 | 7.359 | 7.668 | 6.008 | 3.088 | 3.491 |

- 1247 Samples used

Table B7. Surface PM$_{2.5}$ observations to model comparison, daytime (9:00-18:00) ($\mu$g m$^{-3}$).

|      | OS2.5km | OS1km  | OS1km$^{A9}$ | OS1km$^{B9}$ | OS2.5km$^{B9}$ | OS1km$^{B49}$ |
|------|---------|--------|--------------|--------------|----------------|---------------|
| IoA  | 0.372   | 0.356  | 0.364        | 0.495        | 0.555          | 0.625         |
| r    | 0.232   | 0.244  | 0.245        | 0.350        | 0.387          | 0.493         |
| NMGE | 0.816   | 0.837  | 0.827        | 0.657        | 0.579          | 0.487         |
| MGE  | 5.470   | 5.608  | 5.542        | 4.402        | 3.879          | 3.266         |
| CoE  | -0.256  | -0.288 | -0.272       | -0.011       | 0.109          | 0.250         |
| RMSE | 9.607   | 10.312 | 10.034       | 8.059        | 7.286          | 6.626         |
| NMB  | -0.189  | -0.152 | -0.166       | -0.231       | -0.281         | -0.258        |
| MB   | -1.264  | -1.016 | -1.109       | -1.546       | -1.881         | -1.726        |

- 1862 Samples used

Table B8. Surface PM$_{2.5}$ observations to model comparison, nighttime (18:00 to 9:00) ($\mu$g m$^{-3}$)

|      | OS2.5km | OS1km  | OS1km$^{A9}$ | OS1km$^{B9}$ | OS2.5km$^{B9}$ | OS1km$^{B49}$ |
|------|---------|--------|--------------|--------------|----------------|---------------|
| IoA  | 0.193   | 0.170  | 0.173        | 0.337        | 0.471          | 0.528         |
| r    | 0.163   | 0.183  | 0.178        | 0.277        | 0.368          | 0.442         |
| NMGE | 0.782   | 0.804  | 0.801        | 0.642        | 0.512          | 0.457         |
| MGE  | 5.313   | 5.466  | 5.444        | 4.367        | 3.483          | 3.105         |
| CoE  | -0.614  | -0.660 | -0.653       | -0.326       | -0.058         | 0.057         |
| RMSE | 7.467   | 7.841  | 7.834        | 6.542        | 5.373          | 5.032         |
| NMB  | -0.293  | -0.302 | -0.293       | -0.309       | -0.293         | -0.294        |
| MB   | -1.992  | -2.050 | -1.989       | -2.098       | -1.991         | -1.995        |

- Samples used