# Peer review of "An Evaluation of the Efficacy of Very High Resolution Air-Quality"

_Atmospheric Chemistry and Physics, 2018_

## Referee Comment (RC2) · Anonymous Referee #2 · 28 Jan 2019

The authors present an interesting and well-planned study to evaluate the impacts of high-resolution modeling on air quality model performance. As growing computational resources facilitate model runs at higher grid resolutions, it is important to understand the extent of the improvements that can be expected from increased resolution and limitations that will continue to constrain model performance, especially if tied to traditional assessment metrics. For this reason, the study conveys a valuable message that should be shared with the modeling community. The study is carefully structured and the manuscript is well-written. However, several modifications can be made to strengthen the manuscript. Some comments are included below:

[Figure]

1. While the manuscript compares model performances under 2.5 and 1 km grid resolutions, the authors should also discuss model performance relative to acceptable performance benchmarks for air quality modeling. Do the simulations, with either grid resolution, meet recommended performance benchmarks, for example those reported in Emery, et al., 2017 (doi:10.1080/10962247.2016.1265027)? Showing that the modeling was able to meet standard performance expectations would add confidence to the conclusions drawn about the effect of increasing resolution by indicating that the case is an adequate one to draw conclusions from.

2. The manuscript describes 1km grid simulation as "very high resolution". However, recent work with regional-scale models such as CMAQ or WRF-Chem has been carried out at horizontal grid resolutions of 1 to 3 km. Many of the modeling studies referenced in the manuscript are several years old. A deeper discussion of the progression and current state of grid resolution in Eulerian air quality modeling would strengthen the paper. The paper should discuss what constitutes "very high resolution" at present and, more importantly, what maximum level of resolution can be expected from existing modeling frameworks given the dependence of existing subgrid-scale parameterizations on grid resolution.

3. Although the manuscript's analysis is well structured, some additional discussion of how the findings can be expected to be representative of air quality modeling beyond this specific simulation would be beneficial. Do the authors expect the findings to remain consistent across often applied increasing resolution levels in regional-scale air quality modeling, for example 36km to 12km to 4 km? Should similar conclusions be expected over more urban domains?

4. The manuscript states that the study results are "strongly suggestive of the presence of issues such as illustrated in Figure 3", that is plume structures that are better represented by the higher resolution but more affected by errors in wind fields. An illustrative example of this taken from the simulated results would strengthen this conclusion. A comparison of simulated plumes that mirrors the schematic included in figure 3 would

be beneficial.

5. The authors briefly mention the connection between grid resolution in air quality modeling and associated health impacts projections (line 67-70). Previous work has looked at the impact of increasing grid resolution and improved model performance on health effects estimates, and how these sources of uncertainty compare (e.g., Thompson, et al., 2012, doi:10.5194/acp-12-9753-2012). Some additional discussion of the role of uncertainty due to grid resolution in the larger context of air quality impact assessments, including exposure and health impacts, would be beneficial.

Smaller comments:

- Lines 64-65: This sentence is unclear.

- Lines 97-99: Expand on this statement. What specifically makes the VHR representations more realistic?

- Line 231: Remove "for areas"

- Line 462: Changing "first three columns" to "third column", might be clearer

- Tables 2-5: Including the definition of each acronym used for the metrics somewhere on the chart or at the beginning of the charts would improve readability.

- Line 567-569: The issue of air quality models excessively mixing pollutants along the vertical dimension within the boundary layer has been previously acknowledged by several studies (e.g. Garcia-Menendez, et al., 2014, doi:10.1016/j.scitotenv.2014.05.108).

- Figure 8 needs to be improved. The x-axis of the left panels is illegible. Lines and colors on the right plots are a bit hard to observe as well; a higher resolution/quality plot would help.
* * *

---

## Author Comment (AC1) · 6 Mar 2019

Our responses to the Referee comments appear in the attached PDF file, as well as being supplied below (the latter, created in Word, includes formatting for easier reading).

Response to Referree's comments for "An Evaluation of the Effiᄃcacy of Very High Resolution Air-Quality Modelling over the Athabasca Oil Sands Region, Alberta, Canada", Matthew Russell et al, submitted to ACP. March 6, 2019.

Anonymous Referee 1: Comment: "This paper carried out a series of model-to-model

experiments to evaluate the model performances of simulations at 2.5km and 1km grid-cell, respectively. The comparisons and evaluations between the simulations at 2.5km and 1km grid-cell with surface and aircraft observations were also presented. The simulation results indicate that the 1km model capture simulation values closer to observations than the 2.5km model, while, in general, the 1km simulation has worse performance than the 2.5km simulation. The study is a valuable work, and the design of the simulation experiment is rigorous and reasonable. It suggests that using a smaller grid cell size to get better model results is unnecessary, since it won't always lead to simulation improvements, while other factors such as meteorology and emission should be taken into account to improve the model simulation performance. The paper is well written and clearly structured."

We thank the reviewer for these positive comments.

Comment: "The paper could be further improved if the following comments were addressed:

1. Introduction: when talking about the previous research results, most of the cited papers were published before 2013. Adding some latest research findings might be better and background for other models such as CMAQ model (1. Eder, B., and S. Yu, 2006. A performance evaluation of the 2004 release of Models-3 CMAQ. Atmospheric Environment, 40: 4811-4824; 2. S. Yu et al., 2014. Aerosol indirect effect on the grid-scale clouds in the two-way coupled WRF-CMAQ: model description, development, evaluation and regional analysis. Atmos. Chem. Phys. 14, 11247–11285, doi:10.5194/acp-14-1-2014 )."

We have added a reference to a meta-analysis of a large number of air-quality models (Emery et al., 2017) in the revised manuscript, since the latter deals in a broader sense with model performance evaluation. Eder and Yu 2006 does not deal with the issue at hand in our paper; the impact of model resolution on model performance, and therefore should not be referenced here. We have included the reference to Yu et al 2014 in

the revised Introduction, however, since it deals with two resolutions (12km and 4km), although more in the sense of the different microphysics parameterizations that must be employed going between those two scales, rather than the impact of resolution within a single scale for which the same microphysics is being used, as carried out in our work.

Comment: "2. Page 7, 2.1.GEM-MACH: The authors need give a detailed description about how "the the physics module of the Global Environmental Multiscale meteorological forecast model" was used for different horizontal grid cell size. Specially, for grid cell size >4 km, you will need both resolve cloud and subgrid cloud scheme, while for grid cell size <4 km such as 2.5 km or 1 km, you will need turn off the subgrid cloud scheme in your simulations. Otherwise, you will double count the subgrid cloud effect. This is one of the biggest problems for this study. You must clearly describe these physics module in your paper. This may help you to understand your results."

We have modified the text to make it clear that the two resolutions compared (2.5km and 1km) both use only the Milbrandt-Yau (MY) double moment scheme for the cloud microphysics. We also discuss the cloud parameterizations used at the larger scales (note that these are not cross-compared in the paper). The MY scheme was also used in the intermediate 10km resolution simulation before the explicit microphysics scale was reached – this reduces the spin-up time required for the hydrometeors in the explicit microphysics scheme employed at both resolutions compared in the paper. The following text was added/modified:

"Four levels of nesting have been employed in our simulations, shown in Error! Reference source not found.(a). This version of GEM-MACH operates on a rotated latitude-longitude coordinate system wherein the position of the coordinate system poles are set by the user, allowing rotations of the grid with decreasing grid cell size during nesting. The outermost nested grid corresponds to the westernmost two-thirds of the operational GEM-MACH forecasting domain, with a 10km grid cell size, and employs a combination of the Kain-Fritsch sub-gridscale convective cloud scheme (Kain and

Fritsch, 1990; Kain, 2004) and a Sunqvist (1988) for cloud parameterizations. Within that outer grid is nested a 10 km grid cell size western Canada domain (yellow region, Error! Reference source not found.(a)) which has been rotated to match the horizontal orientation of the Rocky Mountains, and which makes use of a double-moment microphysics scheme (Milbrandt and Yau, 2005a,b) in place of the Sundqvist (1988) parameterization. The intention of this intermediate local 10km simulation domain was to provide initial hydrometeors for the two innermost domains, to reduce the "spin-up" time required for the inner domains' meteorology to reach an equilibrium with respect to cloud formation. The latter two domains (2.5km and 1km grid cell sizes) resolve the cloud microphysics explicitly using the double moment scheme alone and no convective parameterization (Milbrandt and Yau, 2005a,b). "

So both 2.5km and 1km resolution are using the same double-moment scheme for their simulations; there is no difference in the cloud microphysics approach used for the model grid cell sizes compared in the paper.

Comment: "3. P8, L200-201: "The forecasts run in a repeating cycle from new meteorological analyses on every 36 hours, and hence are constrained by observations: ". How can the forecasts be constrained by the observations? I believe that you are doing retrospective simulations other than forecasts. Please explain it clearly."

We have clarified the manuscript on this point; the intent was to point out how meteorological analyses with data assimilation underlie the meteorological initial and boundary conditions in our simulations. There are two main considerations: (1) The model simulations were carried out using a configuration of models which mimics an operational forecasting production. (2) In the latter, meteorological observations are utilized in each successive forecast cycle to improve the model initial meteorological state for any given forecast. These same data-assimilated "analysis" fields are the starting points for our model cycling strategy. The use of data assimilation in weather forecasting has been in place in weather forecasting for many years. We have modified our description as follows:

[Figure]

"Model simulations mimic an operational forecasting system, starting from the use of archived, data-assimilated meteorological analyses as meteorological input and boundary conditions every 36 hours. The use of analysis fields is a standard meteorological forecasting practice to prevent the chaotic drift of the model results from observed meteorology over time. The outermost 10km domain uses initial and boundary conditions from the output of a meteorological simulation, that is itself driven by an analysis field. The outermost domain model then carries out a 36-hour forecast, of which the first 6 hours are discarded as spin-up; the final 30 hours are used as initial and boundary conditions for the rotated 10 km grid cell size domain (the OS10km domain). An OS10km simulation of 30 hours is then carried out. . ."

Comment: "4. P14, regarding the model evaluation metrics, you can refer to the following paper for the definitions these metrics (S. Yu et al., 2006. New unbiased symmetric metrics for evaluation of air quality models. Atmospheric Science Letter, 7, 26-34)."

We had already provided the reference for the metrics in our original manuscript – the "openair" package of Carslaw and Ropkins, 2012. We have added an additional sentence at the same point in the text, "Further discussion of different metrics for model evaluation may also be found in Yu et al., (2006)." The Index of Agreement described in Carslaw and Ropkins is apparently another type of symmetric metric as described in Yu et al. (2006)

Comment: "5. Page 16, line 394: "compare" => "comparing""

Done.

Comment: "6. Page 17, line 418: "Selecting of " => "Selecting""

The original in the manuscript was "Selection of", not "Selecting of"; the original will be used.

Comment: "7. Page 20, line 515-517: "The expected advantages of the small grid cell size, such as better representation of the concentrations of species within plumes and

hence better representation of their reactive chemistry (c.f. Lonsdale et al., 2012), may be lost in a standard performance analysis due to these other issues.". It proposes an important question, is it scientific to evaluate the model performances using the standard statistical metrics? Please have more explanations."

We would not go so far as to characterize the standard performance metrics as in some way less scientific, rather, they specifically evaluate a point-to-point, obs to model comparison. We feel that this is still the most rigorous way to evaluate an air-quality model for performance at specific monitoring locations. However, what we point out in our work is that the additional performance metrics such as those presented in our work show:

(A) That higher resolution model simulations, even if their standard metric performance has not improved, or is even degraded relative to lower resolution models, nevertheless have scientific value. We have shown that the plume magnitudes are better captured in the high resolution models, if not their location.

(B) That the lack of improvement for the standard metrics when going to higher resolution is likely due to difficulties with predicting very localized transport conditions. This is actually a crucial point, in that it suggests that the pathway to improving model performance at higher resolutions should at least in part focus on improving the meteorological transport representation, if needs be through the use of data assimilation of local meteorological observations.

We have emphasized these points in the conclusions with a short additional paragraph as follows:

"Our results should not be taken as an indication that the standard metrics for model comparison are in some way flawed – they provide the most rigorous method for evaluating the performance of a model at specific monitoring locations and specific times. However, the ancillary performance assessment methodology presented here shows that models with very small grid sizes, which may have standard performance metric

scores that have not improved or even have degraded relative to larger grid cell size models, nevertheless have scientific value, in terms of being better able to capture plume concentrations and hence plume chemistry, if not plume position. The work also suggests that the prediction accuracy of very local transport conditions may be a large factor in preventing the smaller grid cell size models from achieving improved performance in standard performance analyses."

Comment: "8. Page 23, line 543-547: "We also noted substantial differences in the day and night performance of both models across the methodologies." How to explain the differences?"

We have added the following sentences: "The study area is located in a broad river valley with frequent slope-defined anabatic/akatabic and drainage flow events. These often have a diurnal nature, and may explain part of the day/night differences. Example sources of these differences may include the relative ability of the driving meteorological model to capture daytime versus nighttime mixed layer turbulence and planetary boundary layer height."

Comment: "9. Table 2 to table 7, and tables in appendix: "GME" => "MGE" and "NGME"=> "NMGE"."

Corrected.

Comment: "10. Page 28, line 613: "Decrease to"=> "Decreasing"."

Done.

Comment: "11. Page 28, line 620: "this"=> "the"?. "

We feel that "Despite this difficulty" is better in this context; the paragraph thus references the one immediately before it.

Comment: "12. Acknowledgements, line 635: "with"=> "wish" or "want"."

Done (used "wish").

Anonymous Referee 2:

Comment: "The authors present an interesting and well-planned study to evaluate the impacts of high-resolution modeling on air quality model performance. As growing computational resources facilitate model runs at higher grid resolutions, it is important to understand the extent of the improvements that can be expected from increased resolution and limitations that will continue to constrain model performance, especially if tied to traditional assessment metrics. For this reason, the study conveys a valuable message that should be shared with the modeling community. The study is carefully structured and the manuscript is well-written."

We thank the reviewer for these positive comments.

Comment: "However, several modifications can be made to strengthen the manuscript. Some comments are included below:

1. While the manuscript compares model performances under 2.5 and 1 km grid resolutions, the authors should also discuss model performance relative to acceptable performance benchmarks for air quality modeling. Do the simulations, with either grid resolution, meet recommended performance benchmarks, for example those reported in Emery, et al., 2017 (doi:10.1080/10962247.2016.1265027)? Showing that the modeling was able to meet standard performance expectations would add confidence to the conclusions drawn about the effect of increasing resolution by indicating that the case is an adequate one to draw conclusions from."

We appreciate the importance and utility of carrying out a meta-analysis of multiple air-quality studies in order to report the current "state of the science" using common air-quality metrics, and of providing a relative ranking of model results at the time of reporting, as was done in Emery et al. (2017). We have added the above reference to our Introduction with that thought in mind. However, we respectfully disagree with the reviewer's contention that the ability (or failure) for a given modelling study to fall within the percentiles reported in Emery et al. (2017) impacts the confidence of conclusions

drawn in a diagnostic study of relative model performance with respect to a specific parameterization or algorithm employed. Emery et al. (2017) state that "The purpose of benchmarks is to understand how good or poor the results are relative to historical model applications of similar nature, and to guide model performance improvements prior to using the model results for policy assessments. To that end, it also remains critical to evaluate all aspects of the model via diagnostic and dynamic methods." Our study is an example of Emery et al.'s definition of a "diagnostic" evaluation, "in which chemical and physical processes within the model are analyzed individually and collectively". The work of Emery et al. (2017) provides a useful evaluation of the range of model performance for specific chemical species at the time of writing (specifically, hourly or maximum daily average 8 hour $O_3$, 24 hour average PM2.5 and speciated PM2.5). Their results are useful in the context of operational model evaluation, but they do not add or reduce confidence in diagnostic evaluations such as carried out in our work. Rather, the two approaches are separate, albeit complementary, avenues to identify model performance issues. The comparison of a model score to other models (in other domains, with other emissions data, etc.) allows a ranking relative to past performance. It will not, if that performance is either improved or degraded relative to the historical simulations, necessarily explain the reasons why this may be the case, unless a diagnostic evaluation, such as the one we have carried out, is employed. Diagnostic evaluations such as our model grid cell size comparison, however, provide that guidance. With this in mind, we have added the following sentences to the Introduction, "The current state of model science is typically evaluated through multi-model intercomparisons (e.g. Im et al., 2015), and the meta-analysis of these studies can be used to provide useful benchmarks to assess current model performance for specific model species and observations (Emery et al., 2017). However, such studies do not identify the causes for good or poor performance relative to the benchmarks – diagnostic studies, "in which chemical and physical processes within the model are analyzed individually and collectively" (Emery et al., 2017) are required for this purpose. Examinations of the impact of model grid cell size on performance are an example of such a

diagnostic evaluation."

Comment: "2. The manuscript describes 1km grid simulation as "very high resolution". However, recent work with regional-scale models such as CMAQ or WRF-Chem has been carried out at horizontal grid resolutions of 1 to 3 km. Many of the modeling studies referenced in the manuscript are several years old. A deeper discussion of the progression and current state of grid resolution in Eulerian air quality modeling would strengthen the paper. The paper should discuss what constitutes "very high resolution" at present and, more importantly, what maximum level of resolution can be expected from existing modeling frameworks given the dependence of existing subgrid-scale parameterizations on grid resolution."

We have added the following text to the Introduction:

"Air-quality model grid-cell size typically follows the grid-cell sizes used in weather forecasting models, which have followed a gradual progression towards finer discretization where more explicit representation of cloud formation and local radiative transfer effects may be represented. The most recent weather forecasting applications (e.g. Leroyer et al., 2014) have reached grid-cell sizes as small as 250m over limited domains such as individual cities, and have shown promising results in terms of being able to resolve some aspects of local circulation. In addition, as grid resolution reaches the 3 to 4 km scale, explicit cloud microphysics packages may be used, allowing potentially better performance, particularly with regards to feedbacks between meteorology and chemistry (Yu et al., 2014; Gong et al., 2015). However, while these models promise better physical representation of local chemistry, their performance may be limited by the quantity and availability of initialization and boundary condition meteorological data; these data may be used in a data assimilation context to improve their initial state. The accuracy of broader-scale meteorological predictions may thus influence local model accuracy, despite the ongoing reduction in meteorological model (and consequently air-quality model) grid cell size. Some recent air-quality model simulation studies with grid cell sizes on the order of one to four km include Thompson and Selin (2012), Li et

al. (2014), Joe et al. (2014), Kheirbek et al. (2014), Kheirbek et al. (2016), and Pan et al., (2017).

Comment: "3. Although the manuscript's analysis is well structured, some additional discussion of how the findings can be expected to be representative of air quality modeling beyond this specific simulation would be beneficial. Do the authors expect the findings to remain consistent across often applied increasing resolution levels in regional-scale air quality modeling, for example 36km to 12km to 4 km? Should similar conclusions be expected over more urban domains?"

This is a very good question, though difficult to answer quantitatively without broadening the scope of our original study significantly. We have added the following paragraph as a short Discussion section before the Conclusions, to try to address the issue in qualitative sense:

"A key result of our current work is that 1km grid cell size simulations resulted in improved prediction of plume concentration maxima relative to 2.5km grid cell size simulations, despite having no improvement using standard scoring methodologies. We also have described a scoring approach wherein these potential advantages of higher resolution may be quantified. We believe that flow field effects such as described in Figure 3 are a general result of increasing grid resolution, but note important caveats, which include:

(1) The availability of meteorological observation and high resolution emissions data to provide model driving information, and the resolution and proximity of this information to the simulation location. Both will influence the relative importance of grid cell size on model results. If this information is available in a higher resolution than the lower of two grid cell size simulations being compared, and/or is used via data assimilation to improve model initial meteorological conditions, our expectation is that the smaller grid cell size model may outscore the larger grid cell size model, even for more standard metrics.

[Figure]

(2) The extent to which local, versus synoptic, weather conditions drive flow in a given region. For example, in the urban heat island meteorological simulations of Leroyer et al. (2014), the accuracy of local flow predictions was shown to be extremely dependent on the representation of the urban heat island, and the accuracy of the latter was critically dependent on the grid cell size (which in this example went down to 250 m). In this respect, for meteorological conditions wherein local factors can dominate the flow, and where those conditions may be adequately modelled only at very high resolution, we would again expect the smaller grid cell size simulation to provide better performance, for standard metrics.

(3) Conversely, model performance using standard metrics should not be expected to increase with successively larger and larger grid sizes; the accuracy of even the synoptic flow field will not be captured as model resolution decreases. Given these considerations, we recommend that modellers should attempt successively smaller grid cell sizes to determine the following: first, the point at which, for their particular system and simulation location, subsequent grid cell size reductions fail to improve performance; and second, to make use of still higher resolutions for studies wherein the point-to-point comparison is less important, and other factors such as accurately capturing the plume chemistry are more crucial."

Comment: "4. The manuscript states that the study results are "strongly suggestive of the presence of issues such as illustrated in Figure 3", that is plume structures that are better represented by the higher resolution but more affected by errors in wind fields. An illustrative example of this taken from the simulated results would strengthen this conclusion. A comparison of simulated plumes that mirrors the schematic included in figure 3 would be beneficial."

We have modified Figure 8 in the manuscript, since it provides such an example, and added the following text to the description of Figure 8 in the revised manuscript:

"Panels (a) and (c) of Figure 8 provide a further example of the kind of situation ref-

erenced in Figure 3; surface monitoring station locations are depicted as grey circles, one of which is identified with a pink arrow. This station lies within the plume at 2.5km resolution (Figure 8(a)), and outside of the plume at 1km resolution (Figure 8(c)). While the plume direction is the same at both scales, that is, the large-scale wind field controls the positioning of the plume axis, the smaller grid cell size simulation places a stronger constraint on the accuracy of the wind field. For example, if the simulated large-scale flow direction was inaccurately predicted by only a few degrees, the plume would not appear in the 1km simulation time series at this location, while registering as present in the 2.5km simulation. Nevertheless, the plume maximum concentration is better captured by the smaller grid cell size simulation (compare maximum values in observed aircraft SO2, Figure 8 (b, d)). The higher resolution simulation may thus more accurately simulate the plume maximum concentration – but not its placement in space, as was hypothesized in Figure 3. "

Comment: "5. The authors briefly mention the connection between grid resolution in air quality modeling and associated health impacts projections (line 67-70). Previous work has looked at the impact of increasing grid resolution and improved model performance on health effects estimates, and how these sources of uncertainty compare (e.g., Thompson, et al., 2012, doi:10.5194/acp-12-9753-2012). Some additional discussion of the role of uncertainty due to grid resolution in the larger context of air quality impact assessments, including exposure and health impacts, would be beneficial."

We thank the reviewer for that reference, which has been added to the revised manuscript. Thompson and Selin (2012) noted that they found that increases in resolution to grid cell sizes below 12 km did not improve their health outcome predictions. This may have been due to the same issues as we have noted in our work: higher spatial resolution does not necessarily guarantee a more accurate prediction at the locations at the observation locations. A lower correlation score at decreasing grid size for example would imply a greater degree of difficulty with linking model output to health effects. However, the studies quoted noted that accurate higher resolution

simulations are nevertheless desired for health studies, due to the need to relate exposure to concentrations on the scale of a few kilometers. Our work helps to explain Thompson and Selin (2012)'s findings, and it points to a possible way to further improve high-resolution model results, through local data assimilation, as we noted in our conclusions. We have modified the given sentence in the Introduction to include this information:

"These studies have often demonstrated that failure to account for higher resolution features may result in mischaracterization of concentrations or health impacts (Isakov et al., 2007), although the capability of current models to provide this information with sufficient accuracy is unclear. One study found that increasing resolution did not change predicted health outcomes, and concluded that "resolution requirements should be assessed on a case-by-case basis" (Thompson and Selin, 2012), while others (e.g. Kheirbek et al. (2014), Kheirbek et al. (2016)) have employed 1km resolution without discussing the impacts of resolution on predicted health outcomes.".

We have also modified extended the paragraph to include a new closing sentence:

"The health studies carried out to date highlight the need for better understanding the underlying controlling factors for model accuracy with decreasing grid cell size."

Comment: "Smaller comments: - Lines 64-65: This sentence is unclear."

The sentence has been modified as follows:

"A number of studies have tried to evaluate the benefits of higher resolution simulations and to quantify the impact of sub-grid variability by using different model grid-cell sizes".

Comment: - Lines 97-99: Expand on this statement. What specifically makes the VHR representations more realistic?"

We have added the following lines of text:

"Salvador et al. (1999) studied the prediction accuracy impacts of meteorological model

grid cell size in a region with complex domain, and found that 2km or smaller grid cell sizes were required to resolve local scale complex terrain flow features, and that daytime vertical advection and predictions of turbulent kinetic energy and potential temperature were influenced by grid cell size. Dore et al. (2012) evaluated air quality model NO2 simulations employing 1, 5 and 50km grid cell sizes against observations, and found the best performance for the 1km simulation, with more physically realistic distributions of reactive nitrogen, attributing this performance gain to more realistically precipitation simulations and emissions inputs for the smallest grid cell size. The availability of high-resolution emissions information may be a limiting factor in improved simulations as grid cell size decreases. Valari and Menut (2008) noted that emissions inaccuracy was the principal cause of noise in small grid cell size simulations conducted for the Paris area, and proposed the use of statistical downscaling in favour of predictive modelling at scales at or below 1km grid cell size. "

Comment: "- Line 231: Remove "for areas""

Done.

Comment: "- Line 462: Changing "first three columns" to "third column", might be clearer"

Done.

Comment: "- Tables 2-5: Including the definition of each acronym used for the metrics somewhere on the chart or at the beginning of the charts would improve readability."

It turned out that the word version of the acronym could fit in the tables, so that has been used.

Comment: "- Line 567-569: The issue of air quality models excessively mixing pollutants along the vertical dimension within the boundary layer has been previously acknowledged by several studies (e.g. Garcia-Menendez, et al., 2014, doi:10.1016/j.scitotenv.2014.05.108)."

Thanks, we have added that reference to this discussion: "Garcia-Menendez et al. (2014) have noted similar results for forest fire plume prediction."

Comment: "- Figure 8 needs to be improved. The x-axis of the left panels is illegible. Lines and colors on the right plots are a bit hard to observe as well; a higher resolution/quality plot would help."

Several improvements were done to Figure 8 in the revised version – we used a smaller time interval for panels (b) and (d) in order to better show the differences in the simulations, and increased the size of the font for the axes of all panels and colour bars. We have also modified panels (a) and (c) to address show the similarities between the simulated concentration fields at each resolution and the hypothesized impact of flow inaccuracies given in Figure 3. In the process of updating this Figure, we also noticed that the 11 UT contour field had been used for panels (a,c) rather than the intended 11

Please also note the supplement to this comment:
https://www.atmos-chem-phys-discuss.net/acp-2018-967/acp-2018-967-AC1-supplement.pdf